# Robust differentiation of human enteroendocrine cells from intestinal stem cells

Daniel Zeve[1,2], Eric Stas[1,11], Joshua de Sousa Casal [3,4,5,11], Prabhath Mannam[1], Wanshu Qi[1], Xiaolei Yin[6,7,10], Sarah Dubois[1,8], Manasvi S. Shah[1,2], Erin P. Syverson[2,3], Sophie Hafner[1], Jeffrey M. Karp[5,7,9], Diana L. Carlone [1,2,9], Jose Ordovas-Montanes [3,4,5,9] & David T. Breault [1,2,9 ✉]

Enteroendocrine (EE) cells are the most abundant hormone-producing cells in humans and are critical regulators of energy homeostasis and gastrointestinal function. Challenges in converting human intestinal stem cells (ISCs) into functional EE cells, ex vivo, have limited progress in elucidating their role in disease pathogenesis and in harnessing their therapeutic potential. To address this, we employed small molecule targeting of the endocannabinoid receptor signaling pathway, JNK, and FOXO1, known to mediate endodermal development and/or hormone production, together with directed differentiation of human ISCs from the duodenum and rectum. We observed marked induction of EE cell differentiation and gut-derived expression and secretion of SST, 5HT, GIP, CCK, GLP-1 and PYY upon treatment with various combinations of three small molecules: rimonabant, SP600125 and AS1842856. Robust differentiation strategies capable of driving human EE cell differentiation is a critical step towards understanding these essential cells and the development of cell-based therapeutics.

[1] Division of Endocrinology, Boston Children's Hospital, Boston, MA 02115, USA. [2] Department of Pediatrics, Harvard Medical School, Boston, MA 02115, USA. [3] Division of Gastroenterology, Hepatology, and Nutrition, Boston Children's Hospital, Boston, MA 02115, USA. [4] Program in Immunology, Harvard Medical School, Boston, MA 02115, USA. [5] Broad Institute of MIT and Harvard, Cambridge, MA 02142, USA. [6] David H. Koch Institute for Integrative Cancer Research, Massachusetts Institute of Technology, Cambridge, MA 02139, USA. [7] Center for Nanomedicine and Division of Engineering in Medicine, Department of Anesthesiology, Perioperative, and Pain Medicine, Brigham and Women's Hospital, Harvard Medical School, Harvard-MIT Division of Health Sciences and Technology, Boston, MA 02115, USA. [8] School of Arts and Sciences, MCPHS University, Boston, MA 02115, USA. [9] Harvard Stem Cell Institute, 7 Divinity Avenue, Cambridge, MA 02138, USA. [10] Present address: Institute for Regenerative Medicine, Shanghai East Hospital, Frontier Science Center for Stem Cell Research, School of Life Sciences and Technology, Tongji University, Shanghai, China. [11] These authors contributed equally: Eric Stas, Joshua de Sousa Casal. ✉email: david.breault@childrens.harvard.edu

Enteroendocrine (EE) cells are found throughout the gastrointestinal (GI) tract and represent the most abundant hormone-producing cell type within mammals. EE cells, as a whole, secrete a large variety of hormones, including glucose-dependent insulinotropic polypeptide (GIP, from K cells), cholecystokinin (CCK, from I cells), serotonin (5HT, from enterochromaffin cells), somatostatin (SST, from D cells), peptide YY (PYY, from L cells), and glucagon-like peptide-1 (GLP-1, from L cells), among others[1,2]. In response to physiological and nutritional cues, EE cells, through the production of these various hormones, are responsible for regulating multiple aspects of GI activity and nutritional homeostasis[1,2]. Because of this, EE cells have been implicated in the pathogenesis of GI diseases such as irritable bowel syndrome and inflammatory bowel disease, as well as metabolic diseases such as type 2 diabetes mellitus[3–5].

Similar to other mature intestinal epithelial cells, EE cells are derived from ISCs, which reside within the crypts of Lieberkuhn[6]. In recent years, much progress has been made understanding the mechanisms underlying ISC self-renewal and differentiation using 3D-organoid culture[7–10]. Using combinations of growth factors and small molecules targeting specific transcriptional regulators and signaling pathways, intestinal organoids can either be maintained predominantly as ISCs or differentiated into mature intestinal cells of either the absorptive or secretory lineages[11,12]. For example, maintenance of ISC self-renewal requires activation of canonical Wnt signaling using WNT3a and R-spondin, suppression of bone morphogenic protein (BMP) signaling using Noggin, and inhibition of p38 MAPK signaling using the small molecule SB202190[9–11]. By altering these pathways along with transcriptional regulators, strategies have begun to emerge to direct ISC differentiation into mature intestinal epithelial cell types[8,9,13].

EE cells are defined by expression of the specific hormone each produces and their location along the GI tract[2,14–16]. These cells also express multiple neuroendocrine secretory proteins, including Chromogranin A (CHGA), some of which are more specific to certain EE cell types than others[14,15]. Further, multiple transcription factors are critical for EE cell differentiation and function, including neurogenin 3 (NEUROG3), neuronal differentiation 1 (NEUROD1), and paired box 4 (PAX4)[1,2,10]. Pancreatic and duodenal homeobox 1 (PDX1) has also been shown to regulate gene expression in a subpopulation of EE cells located within the duodenum[17]. The directed differentiation of human EE cells requires the removal of Wnt ligands and inhibition of Notch signaling but, in contrast to mice, human EE cells do not differentiate in the presence of p38 MAPK inhibitors[9,10,16]. Through the use of an inducible NEUROG3 system, multiple studies have shown induction of the human EE cell lineage, with expression and secretion of multiple hormones in organoid cultures of both the small and large intestine[14,18–21]. However, while strong mRNA expression of human mature EE markers has been observed multiple times using small molecule differentiation protocols[8,9,16,22–24], confirmation using protein expression levels is less frequent[16,22,24], being most thoroughly explored in studies evaluating the roles of short-chain fatty acids[22] and isoxazole-9[24] in human intestinal organoid differentiation and hormone production.

Additional factors and signaling pathways have also been implicated in EE cell formation and function. For example, inhibition of Forkhead box protein O1 (FOXO1), a transcription factor critical for stem cell function and energy homeostasis[25,26], has been associated with upregulation of multiple endocrine-associated transcription factors and hormones, including NEUROG3 and GIP[27–29]. In human intestinal organoids derived from induced pluripotent stem cells, transduction of a dominant-negative FOXO1 yielded increased numbers of CHGA+ cells, less 5HT-positive cells, and no change in GLP-1 or SST-positive cells

when compared to control organoids at 230 days[30]. The endocannabinoid signaling pathway, acting through the cannabinoid receptor type 1 (CB1), has also been shown to regulate EE cell function, with studies showing activation of CB1 inhibiting CCK secretion[31,32]. Additionally, rimonabant (Rim), a highly selective CB1 inverse agonist, has been shown to increase human serum GIP levels[33]. C-Jun N-terminal Kinase (JNK) signaling, which has been implicated in regulating ISCs[34,35], also regulates beta-cell function through its actions on PDX1[36,37], and has been suggested to upregulate EE cell differentiation, with deletion of JNK2 reducing the number of CHGA-positive (CHGA+) cells in mouse organoids[38]. The role of JNK in the regulation of EE cell differentiation and PDX1 expression has not been studied in directed differentiation of human ISCs. Further, BMP4 has been shown to induce the expression of various EE hormones in human intestinal organoids[16]. However, aside from BMP4, how the above factors and pathways impact directed differentiation of EE cells has been largely unstudied.

Here, we establish robust EE cell differentiation protocols for human duodenal organoids (enteroids) and rectal organoids (rectoids) using various combinations of small molecules (rimonabant, SP600125 and AS1842856) suggested to alter the activity of CB1, JNK, and FOXO1, respectively. In enteroids, treatment with Rim and SP600125 (SP) leads to marked induction of *CHGA* expression and a corresponding increase in the number of CHGA + cells, as well as in the expression of multiple hormones, including SST, 5HT, CCK, and GIP. Separately, treatment with AS1842856 (AS) also induced SST, 5HT, and GIP, with an even stronger induction of CHGA+ cells. Comparing levels of hormone secretion, AS leads to a greater induction of 5HT secretion, but less GIP, whereas the combination of Rim and SP (RSP) leads to a greater induction of GIP secretion, but less 5HT. The combination of AS treatment followed by RSP leads to even higher expression of *SST*, and increased secretion of GIP and 5HT, when compared to RSP treatment alone. In rectoids, treatment with base differentiation media (DM) alone induces an increase in the number of CHGA + cells, as well as increased expression, production and secretion of GLP-1 and PYY, EE hormones predominantly produced in the distal GI tract[2]. Together, these robust EE cell differentiation protocols, employing only small molecules, provide important insights into human EE cell differentiation, with potentially important implications for understanding disease pathogenesis and development of cell-based therapeutics.

## Results

**Differentiation media induces *CHGA* expression in human duodenal enteroids**. To optimize human EE cell differentiation, we developed a DM (Supplementary Table 1) by combining specific components from published protocols used to induce EE cells[9,12,16,23], including Wnt3a and two additional small molecules associated with endocrine cell differentiation: (1) betacellulin, a ligand of both EGFR and the ErbB4 receptors[39,40] and (2) PF06260933 (PF), a small molecule inhibitor of MAP4K4[41]. Human enteroids were first cultured in growth media (GM, Supplementary Table 1) for two days, allowing for ISC expansion, followed by DM for 12 days (G2D12). Enteroids maintained for 14 days in GM (G14) were used as controls to assess for changes in gene expression. In contrast to previous strategies aimed at generating EE cells over a five-day period[9,16], exposure to DM for 12 days was compatible with maintenance of enteroid structural integrity (Fig. 1a), possibly due to the continued presence of WNT3a.

To quantify the effectiveness of DM to induce EE cell differentiation, we profiled the gene expression of *CHGA* and

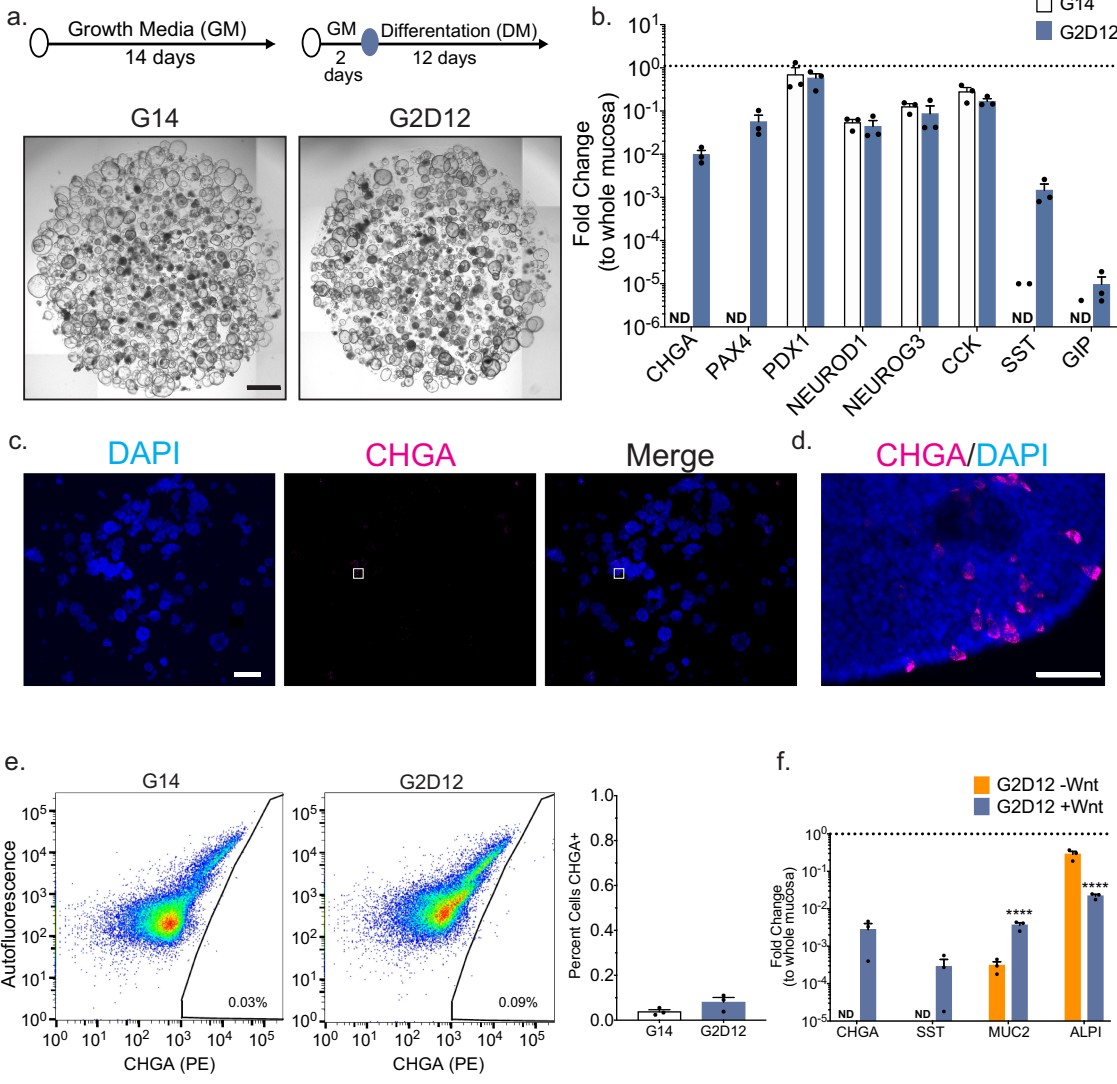

**Fig. 1 Base differentiation media induces CHGA expression in a Wnt-dependent manner. a** Representative light microscopy of enteroids (whole well) grown in growth media (GM) for 14 days (G14) or two days in GM followed by 12 days in differentiation media (DM, G2D12). Specific culture schematic located above each panel, respectively. Scale bar = 1 mm. **b** qPCR analysis of enteroendocrine (EE) markers of enteroids grown in either G14 or G2D12 compared to duodenal whole mucosa and normalized to 18S. Dotted line denotes expression level in whole duodenal mucosa. Representative experiment showing $n = 3$ wells for each condition from a single enteroid line. CHGA = chromogranin A, PAX4 = paired box 4, PDX1 = pancreatic and duodenal homeobox 1, NEUROD1 = neuronal differentiation 1, NEUROG3 = neurogenin 3, CCK = cholecystokinin, SST = somatostatin, GIP = glucose-dependent insulinotropic peptide, ND = not detectable in one or more samples. (**c** and **d**) Representative immunofluorescence staining of CHGA (magenta) in enteroids (whole well) treated with G2D12. Boxed portion in **c** shown magnified in **d**. DNA (4′,6-diamidino-2-phenylindole (DAPI) blue). Scale bars = 1 mm (**c**) and 50 μm (**d**). **e** Left two panels: Representative flow cytometry plots of CHGA-positive (CHGA+) cells from enteroids grown in either G14 or G2D12. Right panel: Percentage of CHGA + cells per well. Representative experiment showing $n = 3$ wells from each condition from single enteroid line. **f** qPCR analysis of mature intestinal gene markers of enteroids grown in either G2D12 without Wnt (G2D12-Wnt) or with Wnt (G2D12 + Wnt) compared to whole mucosa and normalized to 18S. Dotted line denotes expression level in whole duodenal mucosa. MUC2 = Mucin 2, ALPI = Intestinal alkaline phosphatase. Representative experiment showing $n = 3$ wells from each condition from a single enteroid line. ****$p < 0.0001$. Bars show mean ± SEM; two-way ANOVA with Tukey correction for multiple comparisons (**b**, **f**); two-tailed unpaired $t$ test (**e**). Each experiment repeated with at least three different enteroid lines. Specific conditions were excluded from statistical analysis if the data from one or more samples was labeled as not detectable. Source data are provided as a Source Data file.

other lineage markers, including atonal BHLH transcription factor 1 (ATOH1, secretory progenitor cells), mucin 2 (MUC2, goblet cells), lysozyme (LYZ, Paneth cells), intestinal alkaline phosphatase (ALPI, enterocytes), and leucine-rich repeat-containing G-protein coupled receptor 5 (LGR5, ISCs). In addition, we assessed the expression of transcription factors required for duodenal EE differentiation and function (PDX1, PAX4, NEUROG3, and NEUROD1), as well as hormones secreted from the duodenum

(CCK, SST, and GIP). To allow for relative comparison to native tissue expression levels, total RNA from whole duodenal mucosal biopsies was included in each qPCR analysis. Exposure of enteroids to G2D12 induced consistent expression of CHGA, PAX4, SST, and GIP, with significantly higher levels of ALPI and significantly lower expression of LGR5 when compared to G14 (Fig. 1b and Supplementary Fig 1a). Despite G2D12's ability to induce EE and enterocyte markers when compared to G14, their overall

expression levels remained considerably lower than whole duodenal mucosa. Levels of *PDX1*, *NEUROD1*, *NEUROG3*, *CCK*, *ATOH1*, and *LYZ* were unchanged compared to G14, with *MUC2* showing a trend towards higher expression in G2D12 enteroids (Fig. 1b and Supplementary Fig 1a). Finally, despite induction in *CHGA* mRNA levels in response to G2D12, analysis of CHGA protein using immunofluorescence staining and flow cytometric analysis revealed fewer than 0.1% of all cells to be CHGA +, showing a trend toward higher expression compared to G14 (Fig. 1b–e).

Removal of WNT3a has been shown to aid EE differentiation[9,10,16]. Analysis of enteroids cultured in DM without WNT3a (G2D12-Wnt) revealed undetectable expression levels of *CHGA* and *SST*, reduced *MUC2* expression, and increased *ALPI* expression when compared with enteroids differentiated with WNT3a (G2D12 + Wnt) (Fig. 1f). To determine whether *CHGA* expression was increased at an earlier time-point during the 14-day differentiation protocol, we performed time-course studies and found that *CHGA* expression was undetectable in enteroids cultured in G2D12-Wnt, while those exposed to G2D12 + Wnt showed expression after six days of starting DM (Supplementary Fig 1b). Furthermore, two of the three enteroid lines exposed to G2D12-Wnt showed low total RNA levels around the eighth day of differentiation, consistent with failure to maintain these lines (Supplementary Fig 1c). Therefore, we concluded that the presence of WNT3a in DM is beneficial to sustain enteroids in long-term culture (14 days) and is not detrimental to EE differentiation. We also evaluated the impact of including betacellulin and PF in our DM and found that both factors led to increased expression of EE cell markers when compared with G14 and G2D12 without betacellulin or PF, as well as maintaining *ALPI* levels when compared to G2D12 (Supplementary Fig 1d, e). Together, these data indicate that our differentiation protocol markedly induced expression of some EE cell marker genes (e.g., *CHGA* and *PAX4*), but is not sufficient to induce a significant increase in the number of CHGA + EE cells compared to undifferentiated controls.

**Treatment with rimonabant and SP600125 induces duodenal EE lineage differentiation.** To identify additional strategies that might further induce EE cell differentiation, we focused on small molecules shown to target endocannabinoid signaling and PDX1 activity. First, we utilized Rim, a highly selective CB1 inverse agonist. In parallel, we used the small molecule SP to inhibit JNK signaling, which has been shown to suppress PDX1 activity[36,37].

Separately, both Rim and SP induced expression of multiple EE lineage markers (*CHGA*, *NEUROG3*, *SST*, and *GIP*) when added to DM, with Rim having a much larger effect (Supplementary Fig 2a). Together, RSP yielded even further increases in *SST* and *GIP* expression compared to Rim or SP alone. PDX1 expression was unchanged under all experimental conditions (Supplementary Fig 2a). Based on these results, Rim and SP were used in combination for all subsequent experiments. The addition of RSP to DM maintained overall enteroid structural integrity during the 14-day differentiation protocol (Fig. 2a and Supplementary Fig 2b). Moreover, compared to enteroids grown in G14 and G2D12, treatment with RSP led to the upregulation of multiple EE markers (*CHGA*, *PAX4*, *NEUROD1*, *NEUROG3*, *CCK*, *SST*, and *GIP*) to levels approximating whole duodenal mucosa (Fig. 2b). Other lineage markers were also increased with RSP exposure, including *ATOH1*, *MUC2*, *LYZ*, *ALPI*, and *LGR5* when compared to G14 and *ATOH1*, *MUC2*, and *LGR5* when compared to G2D12 (Supplementary Fig 2c). Immunofluorescence staining for CHGA showed multiple positive cells within individual enteroids (Fig. 2c), with a large majority of enteroids

(83%) containing CHGA + cells (Fig. 2d). By comparison, only 1.0% of enteroids grown in G2D12 were CHGA+ (Fig. 2d). Quantitative flow cytometric analysis revealed 1.4% of all cells treated with RSP were CHGA+, almost seven times the number seen with G2D12 alone (Fig. 2e).

**Treatment with AS1842856 induces duodenal EE lineage differentiation.** Next, we assessed the impact of adding AS, a well-described FOXO1 inhibitor[42], to our differentiation protocol. Addition of AS led to the formation of small, spherical enteroids (Fig. 3a and Supplementary Fig 3a). Compared to enteroids grown in G14 and G2D12, AS treatment led to the upregulation of multiple EE markers (*CHGA*, *PAX4*, *PDX1*, *NEUROD1*, *NEUROG3*, *SST*, and *GIP*), many to levels approximating whole duodenal mucosa. The only exception to this was *CCK*, which was significantly lower in AS when compared to G14 (Fig. 3b). In addition, AS increased expression of other secretory and ISC markers, including *ATOH1*, *MUC2*, and *LGR5*, when compared to G14 and G2D12 (Supplementary Fig 3b). Interestingly, the use of AS reduced expression of the enterocyte marker *ALPI* when compared to G14 and G2D12, suggesting a possible role for FOXO1 inhibition in the induction of the secretory lineage (Supplementary Fig 3b). Immunofluorescence staining revealed CHGA + cells within a large majority of individual enteroids (85%) (Fig. 3c, d). By comparison, only 3.5% of enteroids grown in G2D12 had CHGA + cells (Fig. 3d). Quantitative flow cytometric analysis revealed 5.2% of all cells exposed to AS to be CHGA +, almost 50 times the number seen with G2D12 alone (Fig. 3e).

**Treatment with rimonabant/SP600125 or AS1842856 promote distinct enteroid differentiation trajectories.** To better assess the transcriptomic distinctions between cells from the various culture conditions, we utilized single-cell RNA sequencing (scRNA-seq) on the 10X platform combined with sample hashing, allowing us to assess the reproducibility of our findings from distinct donors. Enteroids from three individuals were grown in G2D12, RSP or AS, for a total of nine samples. After the 14-day protocol, enteroids were isolated and processed into a single-cell suspension and their condition and patient source were marked using Hashtag antibodies prior to pooling the samples together[43]. After labeling, cells were processed, and single-cell transcriptomes were generated.

By using hashtag labeling, we were able to categorize cells into singlets (significant detection of one antibody), doublets (significant detection of more than one antibody) or negative droplets (no significant detection of any antibodies). Negative droplets were found to have a reduced number of both detected genes and unique genes or unique molecular identifiers (UMIs), supporting the idea that the majority of these datapoints do not correspond to an individual cell (Supplementary Fig 4a). In contrast, both singlets and doublets had appreciable numbers of both genes and UMIs detected. In addition, we found that they had similar UMIs, percentage of mitochondrial RNA, which aids in identification of low-quality cells, and detected gene distributions (Supplementary Fig 4a). However, upon dimensional reduction, we found that singlets and doublets formed distinct groupings in both t-distributed stochastic neighbor embedding visualization (Supplementary Fig 4b), as well as uniform manifold approximation and projection (UMAP) visualization and clustering (Supplementary Fig 4b, c), indicating that although they had similar RNA content, singlets likely correspond to true singlets, whereas doublets correspond to multiple cell types analyzed together in a single droplet.

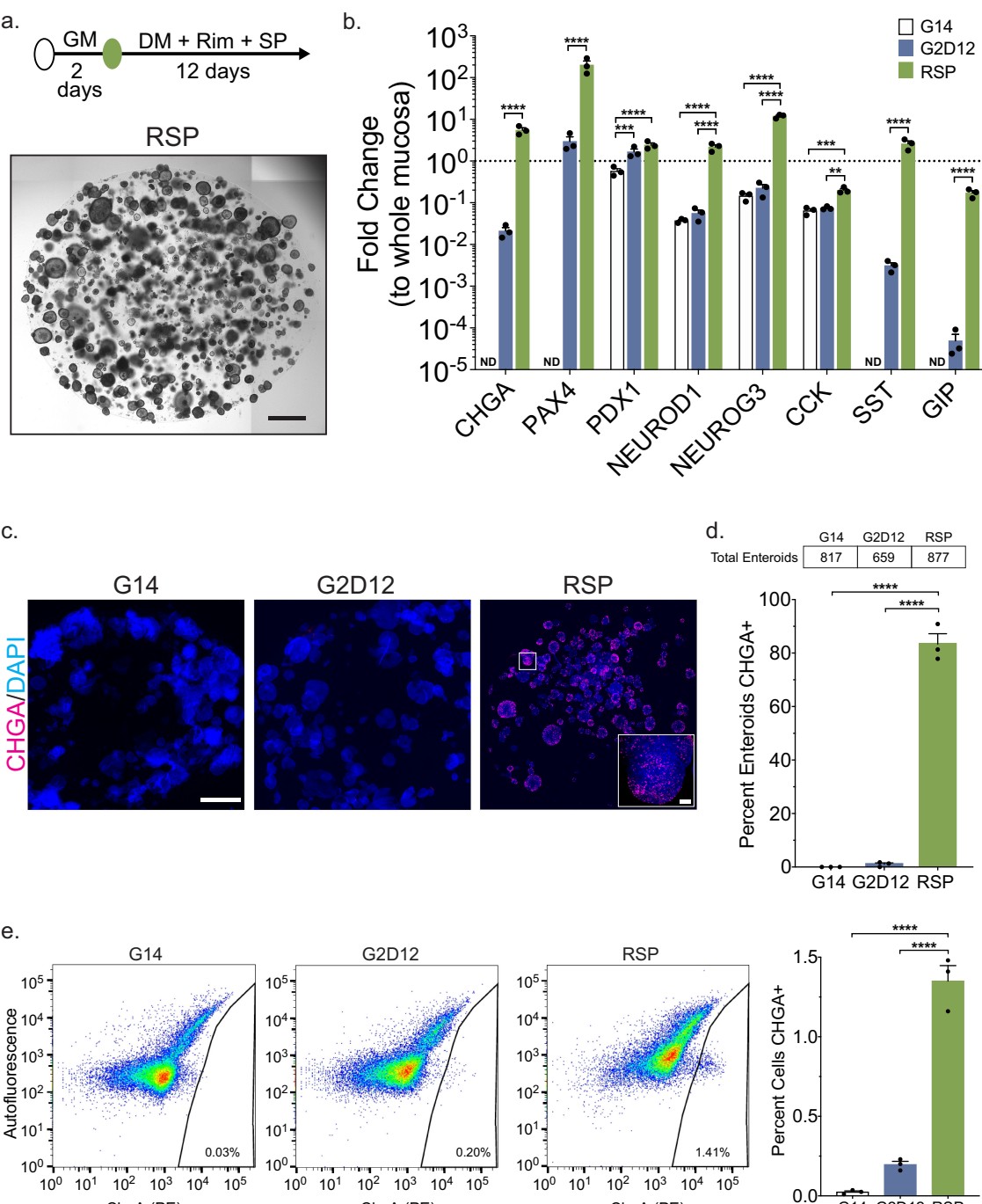

**Fig. 2 Differentiation with small molecules rimonabant and SP600125 induces CHGA expression. a** Representative light microscopy of enteroids (whole well) grown in growth media for 2 days followed by 12 days in differentiation media with rimonabant and SP600125 (RSP). Culture schematic located above panel. Scale bar = 1 mm. **b** qPCR analysis of enteroendocrine markers of enteroids grown in G14, G2D12 or RSP compared to duodenal whole mucosa and normalized to 18S. Dotted line denotes expression level in whole duodenal mucosa. Representative experiment showing $n = 3$ wells from each condition from a single enteroid line. *CHGA* = chromogranin A, *PAX4* = paired box 4, *PDX1* = pancreatic and duodenal homeobox 1, *NEUROD1* = neuronal differentiation 1, *NEUROG3* = neurogenin 3, *CCK* = cholecystokinin, *SST* = somatostatin, *GIP* = glucose-dependent insulinotropic peptide, ND = not detectable in one or more samples. **$p = 0.0025$; ***$p = 0.0009$ (*PDX1*), 0.0003 (*CCK*); ****$p < 0.0001$. **c** Representative immunofluorescence staining of CHGA (magenta) in enteroids (whole well) treated with G14, G2D12 and RSP. Boxed portion magnified in lower right corner. Nuclei (4',6-diamidino-2-phenylindole (DAPI), blue). Scale bars = 1 mm and 50 μm (boxed portion). **d** Percentage of enteroids with positive CHGA staining in G14, G2D12 and RSP treatments. Table above graph shows the total number of enteroids examined per condition. Average results are from three separate experiments from three different enteroid lines or passages. ****$p < 0.0001$. **e** Left three panels: Representative flow cytometry plots of CHGA+ cells from enteroids grown in G14, G2D12, or RSP. Right panel: Percentage of CHGA + cells per well. Representative experiment showing $n = 3$ wells from each condition from a single enteroid line. ****$p < 0.0001$. Bars show mean ± SEM; two-way ANOVA with Tukey correction for multiple comparisons (**b**); one-way ANOVA with Tukey correction for multiple comparisons (**d**, **e**). Each experiment repeated with at least three different enteroid lines. Specific conditions were excluded from statistical analysis if the data from one or more samples was labeled as not detectable. Source data are provided as a Source Data file.

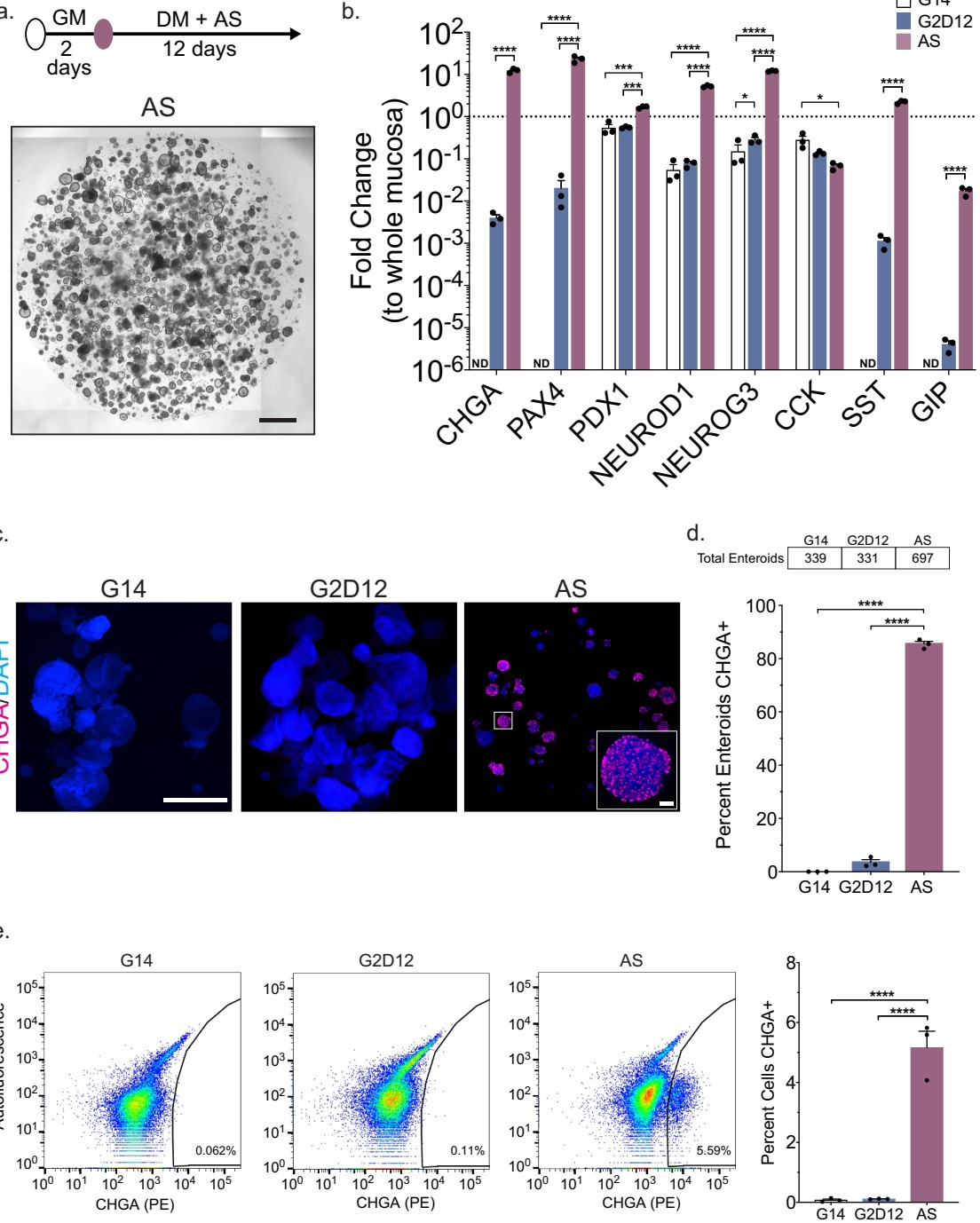

After removal of doublets, negative droplets, and low-quality cells (gene count <200, percentage of mitochondrial RNA > 25%), the resulting dataset consisted of 14,767 cells spread across the three different culture conditions. UMAP visualization revealed distinct cell separation between G2D12 and AS treatment with minimal overlap between the two populations (Fig. 4a). In contrast, enteroid treatment with RSP generated populations that overlapped with the two other culture conditions (Fig. 4a). To further characterize the developing populations within each treatment, we performed Louvain clustering and identified 15 clusters within the combined dataset (Supplementary Fig 4d). Of these 15 clusters, some were composed of cells sourced from all three culture conditions, such as clusters 2, 4, 7, and 9, whereas other clusters were primarily formed by cells from one treatment group, such as cluster 8 (AS) and cluster 3 (G2D12). To assist with manual annotation of each cluster, we calculated the top markers of each Louvain cluster (Supplementary Fig 4e; Wilcoxon rank-sum test, $p_{adj} < 1.38 \times 10^{-21}$ for displayed markers)[44]. By using these markers, as well as classical markers of various epithelial populations[45,46], we broadly classified the data into six populations of epithelial cells, including stem cells, proliferating progenitor cells, progenitor cells, enterocytes, goblet cells, and enteroendocrine cells (Fig. 4b). ISCs were identified by expression of *LGR5, AXIN2* and *ASCL2*[47]. Enterocytes were identified by the expression of *KRT20, FABP2,* and *FABP1*, goblet cells by the expression of *MUC2, FCGBP,* and *GFI1*, and EE cells by expression of *CHGA, NEUROG3, NEUROD1,* and *PAX4*[48,49] (Fig. 4c). Finally, progenitor cells were identified as having a

**Fig. 3 Differentiation with small molecule AS1842856 induces CHGA expression. a** Representative light microscopy of enteroids (whole well) grown in growth media for 2 days followed by 12 days in differentiation media with AS1842856 (AS). Culture schematic located above panel. Scale bar = 1 mm. **b** qPCR analysis of enteroendocrine markers of enteroids grown in G14, G2D12 or AS compared to whole duodenal mucosa and normalized to 18S. Dotted line denotes expression level in whole duodenal mucosa. Representative experiment showing $n = 3$ wells from each condition from a single enteroid line. CHGA = chromogranin A, PAX4 = paired box 4, PDX1 = pancreatic and duodenal homeobox 1, NEUROD1 = neuronal differentiation 1, NEUROG3 = neurogenin 3, CCK = cholecystokinin, SST = somatostatin, GIP = glucose-dependent insulinotropic peptide, ND = not detectable in one or more samples. *$p = 0.038$ (NEUROG3), 0.0130 (CCK); ***$p = 0.0004$ (PDX1, G14 to AS), 0.0009 (PDX1, G2D12 to AS); ****$p < 0.0001$. **c** Representative immunofluorescence staining of CHGA (magenta) in enteroids (whole well) treated with G14, G2D12 and AS. Boxed portion magnified in lower right corner. DNA (4′,6-diamidino-2-phenylindole (DAPI), blue). Scale bars = 1 mm and 50 μm (boxed portion). **d** Percentage of enteroids with positive CHGA staining in G14, G2D12 and AS treatments. Table above graph shows the total number of enteroids examined per condition. Average results are from three separate experiments from three different enteroid lines or passages. ****$p < 0.0001$. **e** Left three panels: Representative flow cytometry plots of CHGA + cells from enteroids grown in G14, G2D12 or AS. Right panel: Percentage of CHGA + cells per well. Representative experiment showing $n = 3$ wells from each condition from a single enteroid line. ****$p < 0.0001$. Bars show mean ± SEM; two-way ANOVA with Tukey correction for multiple comparisons (**b**); one-way ANOVA with Tukey correction for multiple comparisons (**d, e**). Each experiment repeated with at least three different enteroid lines. Specific conditions were excluded from statistical analysis if the data from one or more samples was labeled as not detectable. Source data are provided as a Source Data file.

mixture of stem cell markers, albeit at a lower level than the ISCs, while also expressing mature enterocyte markers. We were able to distinguish proliferating cycling progenitors from other progenitor cells by expression of the proliferation marker MKI67[50] (Fig. 4b, c); however, some progenitor markers were also lowly expressed in secretory cell populations, suggesting that these groups likely also contain secretory progenitors.

To understand how different culture conditions impacted enteroid development and differentiation, we first compared the proportions of the various cell types across each condition (Fig. 4d). We found that treatment with G2D12 primarily gave rise to cells of the absorptive lineage, with 58% identified as enterocytes. The remaining cells exposed to G2D12 consisted of stem cells (5.7%), proliferating progenitor cells (5.4%) and progenitor cells (30%). G2D12 did not give rise to secretory cells, with almost no detection of goblet cells (0.6%) or EE cells (0.2%). In contrast to G2D12, treatment with either RSP or AS led to a decrease in the proportion of enterocytes (RSP: 17%, AS: 2.5%) and an increase in stem cells (RSP: 22%, AS: 42%), goblet cells (RSP: 4.6%, AS: 1.5%), and EE cells (RSP: 4.3%, AS: 18%), with significant differences noted between G2D12 and AS conditions for stem cells, enterocytes, and EE cells.

To further characterize the differentiation path of each culture condition, we used scVelo to model the RNA velocity and differentiation trajectory of each cell type[51]. Briefly, RNA velocity analysis pipelines, such as scVelo, leverage the fact that scRNA-seq can distinguish between un-spliced and spliced variants of each gene. Since recently transcribed RNA is present as an un-spliced variant which is then spliced into its mature form before eventually being degraded, it is possible to calculate the dynamic kinetics of each individual gene's expression across all cells within the dataset based on the ratio of un-spliced to spliced RNA[51,52]. scVelo uses the collection of gene kinetics to infer the direction and speed of cell state changes on a population level. We applied this method to each culture condition and generated vector estimates of cellular transition overlaid on each UMAP (Fig. 4e). In all three conditions, the majority of vector arrows initiate from a stem cell cluster in the lower half of the UMAP. In the G2D12 culture condition, cells primarily move towards proliferating progenitors and progenitor cells, before moving toward the enterocyte cluster. In contrast, in AS treated cells, the vector arrows proceed through a completely different trajectory, with the majority of vectors moving towards the goblet cell cluster and then into the EE cluster. Cells treated with RSP revealed vector arrows in patterns that resembled both G2D12 and AS treated cells, suggesting it may promote early differentiation of both secretory and absorptive cell types. Finally, a small proportion of

vector arrows suggested a "backwards" trajectory, with movement from a differentiated enterocyte into a progenitor population. This behavior may be indicative of de-differentiation or possibly movement towards cell death, as the cells lose features associated with mature cell types[53,54]. Overall, these results indicate that all three culture conditions have distinct differentiation potentials, with AS and RSP giving rise to an increased proportion of EE cells compared to G2D12.

To verify that the EE cells identified in our enteroid system resemble those found in situ, we compared our dataset with a reference gene set from isolated mouse EE cells[45]. This list encompasses 77 genes, including conventional markers such as CHGA, NEUROG3, and the enzyme TPH1 (Supplementary Fig. 4f). An EE cell gene module score was calculated for each cell in our scRNA dataset based on the number of shared features with the reference EE gene set[45], which revealed exclusive marking of cells within our EE cell cluster (Fig. 4f). In addition, when looking across culture conditions, RSP and AS showed an increased effect size in relation to G2D12, with AS having a larger effect size than RSP (Fig. 4g). These results underscore differences observed between the three culture conditions described in previous results and further demonstrate that EE cells generated using these protocols resemble those found within native tissue.

To understand the effect of each culture condition on the differentiation of individual enteroendocrine subsets, we further clustered the EE cells previously identified in our scRNA-seq dataset. Following sub-clustering, we identified a total of 471 EE cells, which consisted of enteroendocrine progenitor cells, enterochromaffin cells and various hormone-producing cells (Supplementary Fig. 5a, b). We defined enteroendocrine progenitor cells by the expression of DLL1 and NEUROG3, and the absence of expression of differentiated markers NEUROD1, CHGA, and ARX, the last of which specifically marks non-enterochromaffin EE cells[15,55] (Supplementary Fig. 5c). We found that RSP treatment gave rise to a greater proportion of enteroendocrine progenitor cells compared to AS (RSP 50% vs AS 13%) (Supplementary Fig. 5d). In contrast, AS treatment led to an increased frequency of mature EE cells compared to RSP (RSP 50% vs AS 87%), with the majority consisting of enterochromaffin cells (RSP 40% vs AS 62% of all cells) (Supplementary Fig. 5d). To investigate the population of peptide hormone-producing cells, we leveraged our scRNA-seq dataset to determine the expression of hormones and related genes that have previously been attributed to various EE subsets[2,15,18,45,56]. Notably, we found expression of both TPH1, associated with enterochromaffin cells, and GHRL, associated with X cells, across both AS and RSP. SST was not differentially

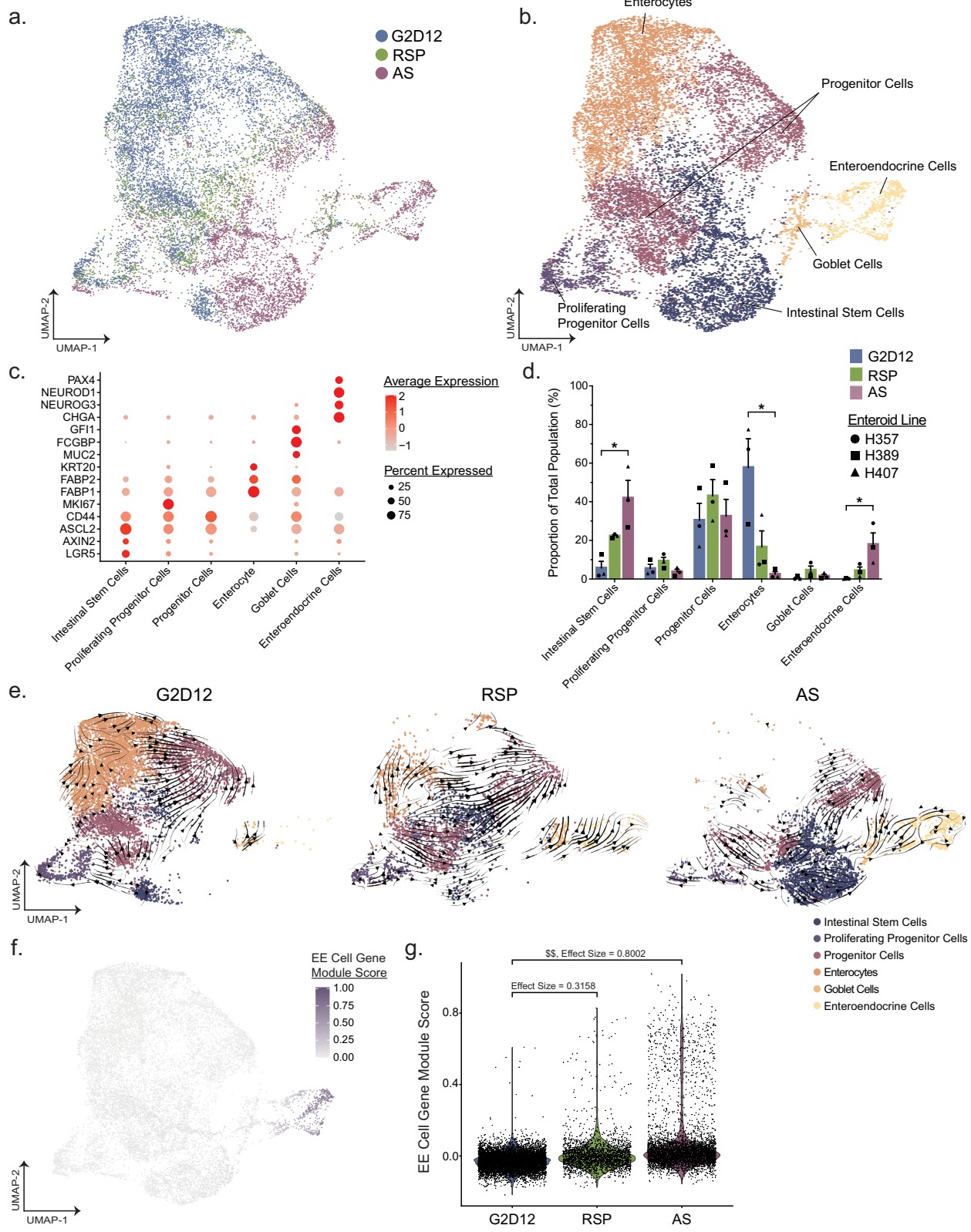

expressed between AS and RSP, while *MLN*, associated with M cells, was identified predominantly in RSP-treated cells, although these markers, along with *GAST*, associated with G cells, *GIP*, and *CCK*, were expressed in only a small number of cells (Supplementary Fig. 5e).

**Treatment with AS1842856 followed by rimonabant and SP600215 increases GIP expression in human duodenal enteroids**. Compared to RSP treated enteroids, AS treatment led to more robust EE differentiation, based on overall induction of multiple EE cell markers and the larger fraction of CHGA + cells

**Fig. 4 scRNA-seq profiling of enteroids cultured with rimonabant/SP600125 or AS1842856. a** Uniform manifold approximation and projection (UMAP) visualization of 14,767 cells summarizing enteroid differentiation from all samples, color labeled by culture condition. **b** UMAP visualization from **a**, color labeled by broad annotated cell identity, following Louvain clustering. **c** Dot plot of the average scaled expression (measured by average Pearson residual) of canonical markers of various intestinal epithelial cell types, plotted against cluster identity. **d** Proportional abundance of epithelial cell subsets by enteroid culture protocol. Each culture condition consists of three different enteroid lines from distinct human donors, as denoted by data point shape. Bars show mean ± SEM; Kruskal-Wallis test with Dunn's post-hoc analysis displayed. *P* values were adjusted using Bonferroni correction for multiple comparisons. *$p = 0.0219$ (Intestinal Stem Cells and Enteroendocrine Cells), 0.0338 (Enterocytes). **e** UMAP visualization of 14,767 cells divided by culture condition and colored by cell identity. Trajectory analysis of each protocol was calculated using scVelo and the vector field was overlaid on top of each UMAP. Arrows represent smoothed averages of the estimated cellular differentiation trajectory, with arrow thickness corresponding to the "speed" of differentiation. **f** UMAP visualization from **a**, with individual cells colored by their EE cell module score. Each score was scaled on a range from 0 to 1. **g** Violin plots of the module score described in **f** split across culture condition. The effect size between culture conditions was calculated as Cohen's *d*; $$0.8 < d < 1.2$. Source data are provided as a Source Data file.

(Figs. 2–4). Direct comparison of the two protocols revealed that AS exposure led to higher expression of *CHGA, NEUROD1, NEUROG3,* and *SST* (Supplementary Fig. 6a), consistent with the higher percentage of CHGA + cells (Figs. 2 and 3). In contrast, we noted that RSP induced higher expression of *GIP* compared to AS (Supplementary Fig. 6a). Given these results, we next hypothesized that the combination of Rim, AS, and SP (RASP) would further increase expression of both *CHGA* and *GIP*. Exposure of enteroids to RASP for the full duration of the differentiation protocol, however, was not compatible with viable enteroids, as evidenced by their irregular morphology and extremely low RNA content (Supplementary Fig. 6b, c).

We next tested the impact of adding RSP after exposure to AS, reasoning that AS treatment would shunt a larger proportion of cells into the EE lineage, as indicated by the scRNA-seq data (Fig. 4d, g), with the later addition of RSP inducing *GIP* expression in more cells than RSP alone. To identify the appropriate time for the addition of RSP, we performed a time-course analysis of AS-treated enteroids and found that multiple transcription factors required for EE differentiation, including *NEUROD1* and *NEUROG3,* showed increased expression around the fourth day of differentiation. Following this, both *CHGA* and *SST* had detectable transcript levels by the sixth day of differentiation (Supplementary Fig. 7a). Given these data, we next hypothesized that AS treatment for six days followed by subsequent exposure to RSP would lead to increased *GIP* expression compared to RSP alone. To test this, we utilized two differentiation strategies: (1) switching from AS to RSP at day six of differentiation (AS→RSP) or (2) adding RSP to AS at day six of differentiation (AS→RASP). Morphologically, AS→RASP produced smaller enteroids compared to other conditions (Supplementary Fig. 7b); however, RNA concentrations were consistently above the minimum threshold of 10 ng/μL, suggesting improved viability over exposure to RASP for the full duration of the differentiation protocol (Supplementary Fig. 7c). Enteroids treated with AS→RSP showed gene expression changes that were similar to RSP alone, aside from *CHGA* and *SST* (Fig. 5a). Enteroids treated with AS→RASP showed 2–3-fold higher gene expression levels of most EE cell markers when compared to AS, with the exception of *PDX1* and *CCK* (Fig. 5a).

Furthermore, exposure to AS for the entire differentiation protocol, i.e., AS and AS→RASP, decreased expression of *MUC2* and *ALPI* and increased expression of *LGR5* compared to those that received RSP alone, for any amount of time (Supplementary Fig. 8a). In terms of other lineages, immunofluorescence staining showed that the enterocyte marker cytokeratin 20 (CK20) was expressed in all differentiation conditions, but with very little stain noted in AS→RASP (Supplementary Fig. 8b). MUC2 staining was only noted in RSP, AS, and AS→RSP enteroids while LYZ, similar to CK20, was present in all differentiation conditions except AS→RASP (Supplementary Fig. 8c, d).

We also evaluated the effect of our differentiation protocols on proliferation and apoptosis after seven and 14 days of culture. As expected, at seven days GM supported cell proliferation, as evidenced by a significantly higher fraction of EdU-positive (EdU+) cells (2.7%) after a 2-h pulse, compared to all other conditions. There were no significant differences between DM only, RSP and AS, which ranged from 0.1 to 0.5% EdU+ cells. At day 14 of culture, no differences in the percentage of EdU+ cells were detected between any groups (which ranged from 0.1 to 0.6%), suggesting that GM conditions are not able to sustain ongoing proliferation indefinitely (Supplementary Fig. 8e). Analysis of apoptosis at seven days showed GM had the lowest percentage of Annexin V-positive cells (13%) compared to each differentiation condition, with RSP showing the highest fraction at 29%. At day 14, despite modest differences between groups, levels of apoptosis were fairly consistent ranging from 19 to 29% (Supplementary Fig. 8f). Further, enteroids exposed to RSP (including RSP, AS→RSP, and AS→RASP) appeared to have more apoptotic cells compared to DM alone and AS.

All four differentiation strategies induced a high fraction of CHGA + enteroids, ranging from 79% to 88% (RSP, 79%; AS, 82%; AS→RSP, 81%; AS→RASP, 88%), as assessed by immunostaining (Fig. 5b, c). Quantitative flow cytometric analysis revealed 3.7% of all cells exposed to AS→RSP to be CHGA+, which was between the 1.0% seen in RSP alone and the 6.1% seen in AS alone. AS→RASP had the highest percentage of CHGA+ cells at 7.1%, more than 100 times the number seen with G2D12 alone (Fig. 5d). Taken together, these data reveal that exposure of human enteroids to AS→RASP is the most effective way to induce EE cell differentiation, in terms of expression of *CHGA, PAX4, NEUROD1,* and *NEUROG3,* as well as overall number of CHGA + cells.

**Hormone production and secretion mirror gene expression changes in duodenal enteroids**. To assess hormone production and secretion during EE cell differentiation, we next assayed for duodenal-associated hormones in response to the various differentiation conditions. Immunofluorescence staining showed that SST was expressed similarly in all differentiation conditions, aside from G2D12, with 42–53% of enteroids containing SST-positive cells (RSP, 45%; AS, 42%; AS→RSP, 53%; AS→RASP, 51%) (Fig. 6a, b). 5HT was expressed in a higher percentage of enteroids treated with AS throughout the entire differentiation period than those that only received RSP (RSP, 44%; AS, 75%; AS→RSP, 59%; AS→RASP, 76%) (Fig. 6c, d). An induction in GIP-positive enteroids was detected in response to all differentiation conditions, with the exceptions of G2D12 and AS→RASP (RSP, 66%; AS, 53%; AS→RSP, 56%; AS→RASP, 8.6%) (Fig. 6e, f). Further, CCK-positive enteroids were predominantly detected in response to RSP (29%) and AS→RSP (30%) compared to all other treatment groups (AS, 4.5%; AS→RASP, 2.6%) (Fig. 6g, h).

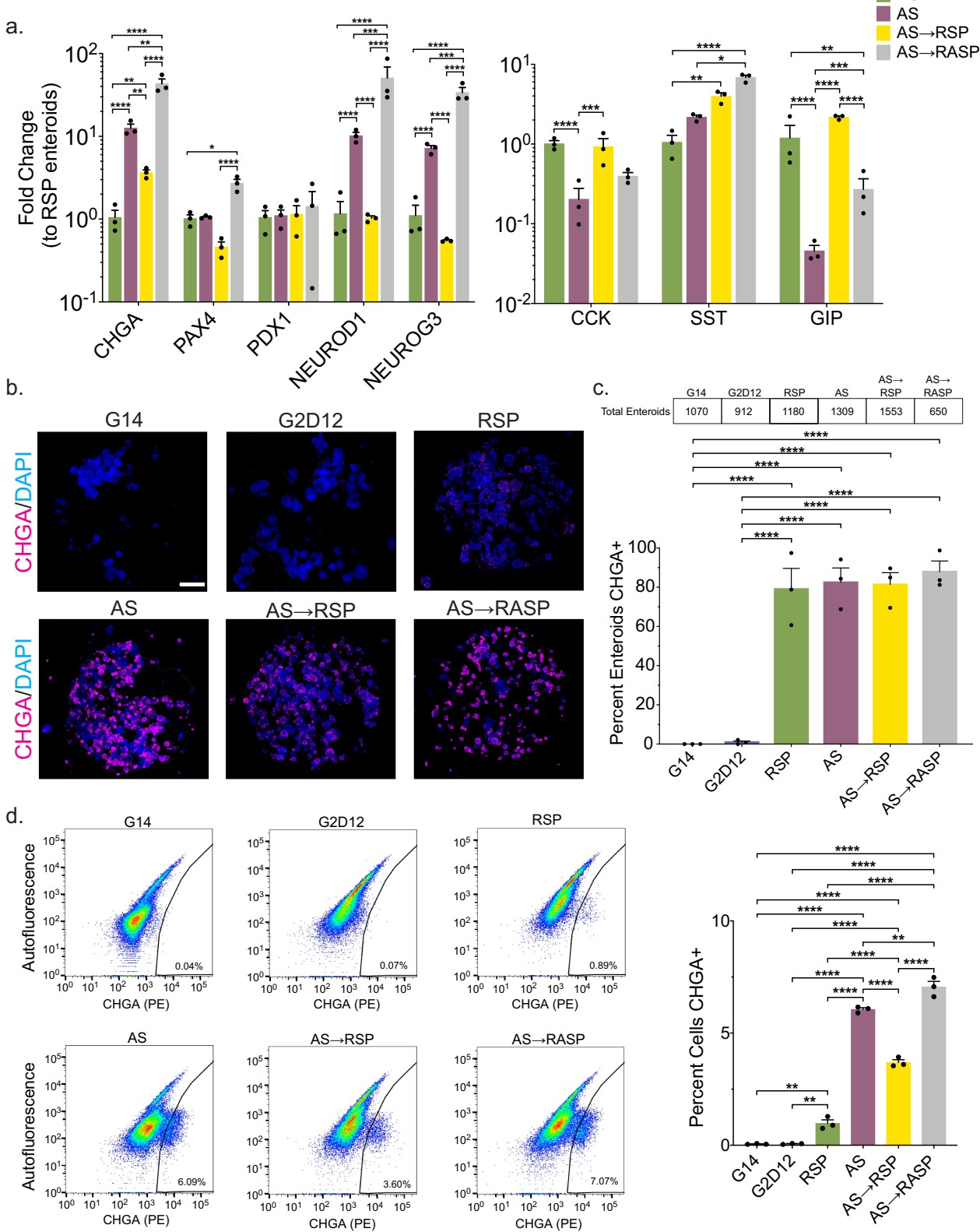

Finally, to assess hormone secretion, we assayed conditioned media from each differentiation condition. Exposure to AS alone induced higher levels of 5HT secretion than all other conditions, with AS→RSP conditioned media showing higher levels of 5HT when compared to RSP and AS→RASP (Fig. 7a). These patterns persisted when controlled for total DNA content, with the exception of a modest difference between AS→RSP and

AS→RASP (Supplementary Fig. 9a). Importantly, secretion of 5HT was significantly increased when differentiated enteroids were exposed to the adenylate cyclase activator forskolin (Fsk) (Fig. 7b and Supplementary Fig. 9b). Secretion of GIP was highest following exposure to AS→RSP, compared to RSP alone, while AS and AS→RASP treated enteroids revealed no secretion of GIP (Fig. 7c and Supplementary Fig. 9c). Secretion of GIP from

**Fig. 5 Combinations of AS1842856 and rimonabant/SP600125 induce different levels of enteroendocrine marker expression. a** qPCR analysis of enteroendocrine markers of enteroids grown in AS, AS for 6 days, followed by RSP only for 6 days (AS→RSP) and AS for 6 days, followed by AS and RSP for 6 days (AS→RASP) compared to enteroids grown in RSP and normalized to 18S. Representative experiment showing $n = 3$ wells from each condition from a single enteroid line. CHGA = chromogranin A, PAX4 = paired box 4, PDX1 = pancreatic and duodenal homeobox 1, NEUROD1 = neuronal differentiation 1, NEUROG3 = neurogenin 3, CCK = cholecystokinin, SST = somatostatin, GIP = glucose-dependent insulinotropic peptide. *$p = 0.0424$ (PAX4), 0.0123 (SST); **$p = 0.039$ (CHGA, RSP to AS→RSP), 0.0059 (CHGA, AS to AS→RSP), 0.0064 (CHGA, AS to AS→RASP), 0.0019 (SST), 0.0010 (GIP); ***$p = 0.0006$ (NEUROD1), 0.0004 (NEUROG3), 0.0003 (CCK), 0.0001 (GIP); ****$p < 0.0001$. **b** Representative immunofluorescence staining of CHGA (magenta) in enteroids (whole well) treated with G14, G2D12, RSP, AS, AS→RSP, and AS→RASP. DNA (4′,6-diamidino-2-phenylindole (DAPI), blue). Scale bar = 1 mm. **c** Percentage of enteroids with positive CHGA staining in G14, G2D12, AS, RSP, AS→RSP, and AS→RASP treatments. Table above graph shows the total number of enteroids examined per condition. Average results are from three separate experiments from three different enteroid lines or passages. ****$p < 0.0001$. **d** Left six panels: Representative flow cytometry plots of CHGA + cells from enteroids grown in G14, G2D12, RSP, AS, AS→RSP, and AS→RASP. Right panel: Percentage of CHGA + cells per well. Representative experiment showing $n = 3$ wells from each condition from a single enteroid line. **$p = 0.0041$ (G14 to RSP), 0.0045 (G2D12 to RSP), 0.0020 (AS to AS→RASP); ****$p < 0.0001$. Bars show mean ± SEM; two-way ANOVA with Tukey correction for multiple comparisons (**a**); one-way ANOVA with Tukey correction for multiple comparisons (**c**, **d**). Each experiment repeated with at least three different enteroid lines. Source data are provided as a Source Data file.

enteroids exposed to AS→RSP was also significantly increased after exposure to Fsk (Fig. 7d and Supplementary Fig. 9d). Overall, these data show that RSP and AS, either alone or in combination, can induce protein expression and secretion of multiple duodenal hormones (5HT, GIP, SST, and CCK) from human enteroids, with specific differences depending on exposure and/or timing of RSP and AS treatment.

**Differentiation media induces EE cell differentiation in human rectoids.** To evaluate the ability of our DM to induce EE cell differentiation outside of the duodenum, we exposed organoids derived from rectal ISCs (rectoids) to the G14 and G2D12 protocols (Supplementary Table 1). Rectoids treated with G14 maintained spherical structures, while those exposed to G2D12 had a mixture of spherical and budding structures (Fig. 8a). In contrast to enteroids, rectoids exposed to G2D12 had significantly increased expression of the majority of EE markers, included CHGA, PAX4, NEUROD1, NEUROG3, SST, GCG, and PYY, many of which were at or above the expression level observed in whole rectal mucosa (Fig. 8b). Notably, rectoids exposed to G2D12 also had significantly higher expression of ATOH1 and MUC2, decreased expression of LGR5, and no change in CAII expression, a marker for absorptive cells of the rectum[57,58], when compared to G14 (Supplementary Fig. 10).

Immunofluorescence staining revealed CHGA + cells within a majority of G2D12 rectoids (72%) (Fig. 8c, d) and quantitative flow cytometric analysis revealed 4.7% of all cells exposed to G2D12 to be CHGA + (Fig. 8e). Additional immunofluorescence staining revealed that both GLP-1 and PYY were expressed in a large number of G2D12 rectoids (GLP-1, 61%; PYY, 40%) (Fig. 9a–d). We next assayed conditioned media from G14 and G2D12 rectoids to evaluate for hormone secretion and found significant levels of both GLP-1 and PYY in G2D12 rectoids compared to G14, which was increased by stimulation with Fsk (Fig. 9e–h, Supplementary Fig 11a–d). Overall, these data indicate that G2D12 is sufficient to induce EE cell differentiation of rectal ISCs.

**Discussion**

Here, we establish that the addition of the small molecules rimonabant, SP600125 and AS1842856 leads to robust differentiation of human EE cells from duodenal ISCs without the use of transgenic modification. This was assessed by EE marker gene expression, which approached levels of the native duodenal mucosa, and protein expression, including immunofluorescent staining, flow cytometric analysis, and hormone secretion assays.

Many of the studies establishing a role for Wnt, Notch, MAPK, and BMP signaling, as well as short chain fatty acids and isoxazole-9, in EE lineage differentiation and hormone production have been performed using the organoid culturing system, which approximates in vivo growth and development due to its 3D nature[8,9,16,22–24]. However, there are significant shortcomings to the analyses of intestinal organoid differentiation that our study has begun to address. First, RNA expression does not always mirror protein expression, as evidenced by the presence of GIP mRNA in AS→RASP enteroids (Fig. 5a) with little protein expression noted on immunofluorescence (Fig. 6e, f) and no secreted protein seen on ELISA (Fig. 7c). Further, our study suggests that enteroids grown in G14 are not appropriate as the sole reference point in differentiation experiments. For example, G2D12, but not G14, treatment induced expression of CHGA mRNA (Fig. 1b), but the comparison of enteroid mRNA expression levels to whole duodenal mucosa revealed that G2D12 alone induces a very low level of CHGA compared to native tissue. Moreover, immunodetection of CHGA expression revealed significant heterogeneity between individual enteroids, even when cultured under the same experimental conditions, as evidenced by only a small minority of enteroids and cells staining CHGA + in G2D12 and the variability in hormone expression using immunofluorescence seen in the other differentiation conditions. These results highlight the limitations of current human EE differentiation protocols.

Multiple human EE differentiation protocols using only small molecules have suggested that removal of WNT3a from the base growth medium is critical for secretory lineage differentiation[9,16]. Our results, in contrast, reveal that removal of WNT3a is detrimental to both EE differentiation and long-term viability, as noted by a lack of CHGA expression at any time during the differentiation protocol and a marked decrease in total RNA levels with time. This is consistent with a previous study suggesting that maintaining high WNT3a concentrations during differentiation was similar, if not better, at inducing CHGA expression than a reduction of WNT3a[23]; however, that study only examined differentiation markers after two days of differentiation. Additional studies have evaluated differentiation of human intestinal organoids using reduced amounts of WNT3a with varying lengths of differentiation and found induction of EE cell markers, when examined[23,24,59,60]. Regardless, our data indicate that high WNT3a concentrations can promote and maintain differentiation of human intestinal organoids for at least 12 days and that removal of WNT3a is not necessary for human EE cell differentiation.

We also show that treatment of human enteroids with rimonabant, an inverse agonist of CB1 signaling, in conjunction with

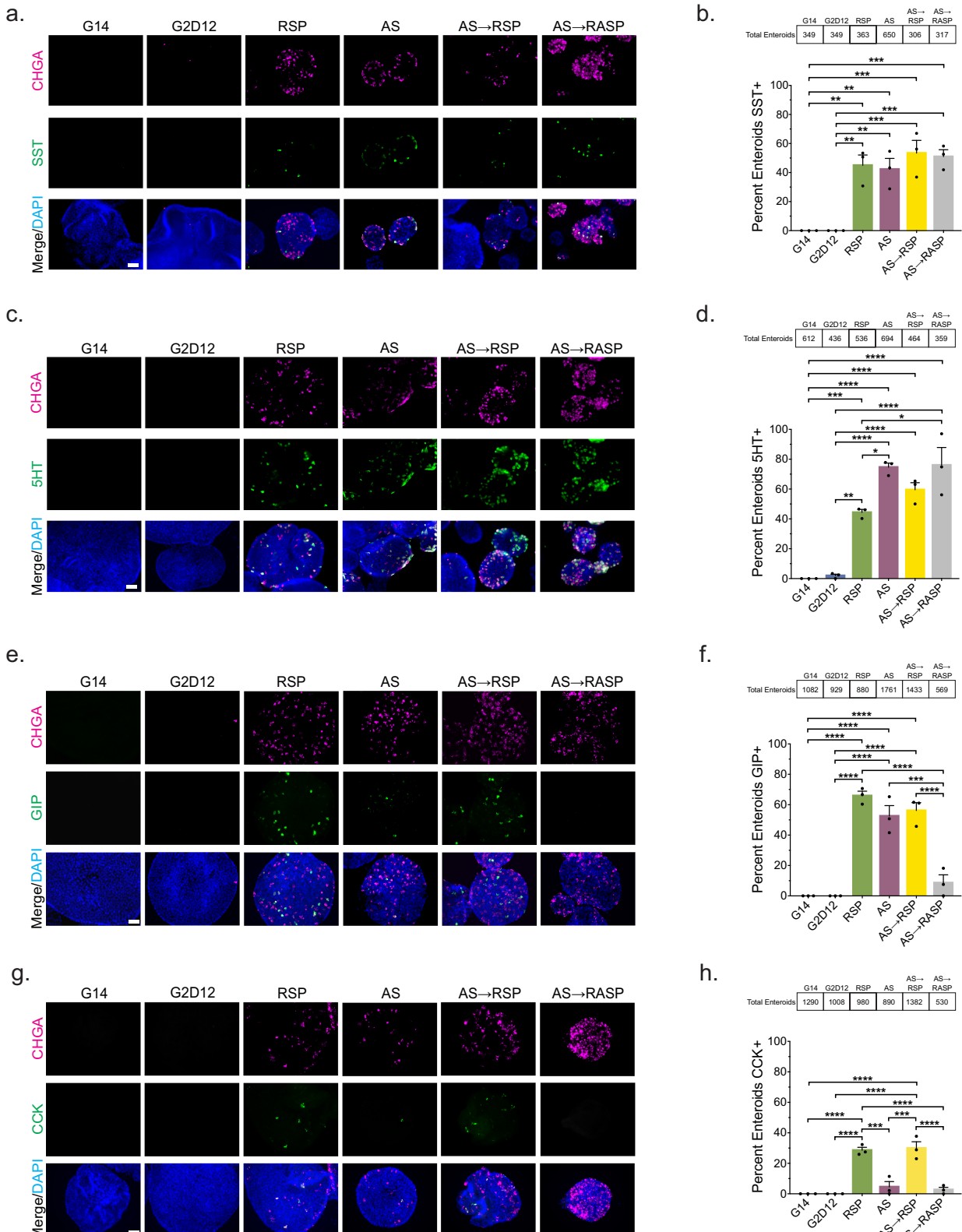

a known JNK inhibitor, led to an increase in EE marker expression and EE cell number, compared with G2D12 alone. We had similar findings when using a well-described inhibitor of FOXO1, which has been previously implicated in beta-cell differentiation[30], but whose role in EE cell differentiation from primary human ISCs has not been closely examined. These protocols showed high levels of human EE cell differentiation compared with prior reports[9,16,23]. While transgenic over-expression of *NEUROG3* generated elevated levels of human EE cells[14,18,19], our methods avoid the use of genetic techniques while maintaining EE marker expression at levels at or above endogenous levels within the human duodenal mucosa. Interestingly, both RSP and AS treatments also led to increased levels of *LGR5* expression, suggesting some induction of stemness. It

**Fig. 6 Multiple differentiation conditions induce hormone production. a** Representative immunofluorescence staining of somatostatin (SST, green) and chromogranin A (CHGA, magenta) in enteroids treated with G14, G2D12, RSP, AS, AS→RSP, and AS→RASP. DNA (4′,6-diamidino-2-phenylindole (DAPI), blue). Scale bar = 50 μm. **b** Percentage of enteroids with SST staining in G14, G2D12, RSP, AS, AS→RSP, and AS→RASP treatments. Average results are from three different enteroid lines or passages. **$p = 0.0017$ (G14 to RSP, G2D12 to RSP), 0.0029 (G14 to AS, G2D12 to AS); ***$p = 0.0004$ (G14 to AS→RSP, G2D12 to AS→RSP), 0.0006 (G14 to AS→RASP, G2D12 to AS→RASP). **c** Representative immunofluorescence staining of serotonin (5HT, green) and CHGA (magenta) in enteroids treated with G14, G2D12, RSP, AS, AS→RSP and AS→RASP. DNA (DAPI, blue). Scale bar = 50 μm. **d** Percentage of enteroids with 5HT staining in G14, G2D12, RSP, AS, AS→RSP and AS→RASP treatments. Average results are from three different enteroid lines or passages. *$p = 0.0183$ (RSP to AS), 0.0134 (RSP to AS→RASP); **$p = 0.0014$; ***$p = 0.0009$; ****$p < 0.0001$. **e** Representative immunofluorescence staining of glucose-dependent insulinotropic peptide (GIP, green) and CHGA (magenta) in enteroids treated with G14, G2D12, RSP, AS, AS→RSP, and AS→RASP. DNA (DAPI, blue). Scale bar = 50 μm. **f** Percentage of enteroids with GIP staining in G14, G2D12, RSP, AS, AS→RSP, and AS→RASP treatments. Average results are from three different enteroid lines or passages. ***$p = 0.0001$; ****$p < 0.0001$. **g** Representative immunofluorescence staining of cholecystokinin (CCK, green) and CHGA (magenta) in enteroids treated with G14, G2D12, RSP, AS, AS→RSP, and AS→RASP. DNA (DAPI, blue). Scale bar = 50 μm. **h** Percentage of enteroids with CCK staining in G14, G2D12, RSP, AS, AS→RSP, and AS→RASP treatments. Average results are from three different enteroid lines or passages. ***$p = 0.0002$ (RSP to AS), 0.0001 (AS to AS→RSP); ****$p < 0.0001$. Bars show mean ± SEM; one-way ANOVA with Tukey correction for multiple comparisons (**b, d, f, h**). Tables above graphs show the total number of enteroids examined per condition. Each experiment repeated with at least three different enteroid lines. Source data are provided as a Source Data file.

has been suggested that EE cells can act as facultative stem cells in the setting of stress and injury[61], which in our organoid system may result from chronic culture and/or loss of Matrigel integrity.

Regarding mechanism, it is not known what downstream changes occur after exposure to Rim. Interestingly, GATA Binding Protein 4 (GATA4) is known to play roles in small intestine formation and in the expression of CCK and GIP, as *Gata4* deletion in vitro and in mice leads to reduced expression of both GIP and CCK[62–65], and GATA4 has been shown in vitro to bind to the GIP promoter[66]. Similarly, both CCK and GIP have been shown to be regulated by endocannabinoid signaling[31–33]. Further, endocannabinoid signaling is known to inhibit cAMP/ protein kinase A signaling[67], which has been shown to play a role in GATA4 activation;[68] therefore, it is possible that reducing CB1 activity could lead to GATA4 activation in human enteroids, leading to ISC differentiation and hormone production. However, in vivo murine studies show that GATA4 is not expressed in mature EE and goblet cells[69,70], suggesting that GATA4 may not be involved in mature EE cell function, but could be involved in EE cell differentiation. In addition, Rim functions as a dual inhibitor of sterol O-acyltransferase 1 (SOAT1 or ACAT1) and SOAT2 (ACAT2)[71], though these play more of a role in the production of steroid, not peptide, hormones[72].

It is notable that two recent scRNA-seq studies analyzing human and murine EE lineage differentiation using inducible expression of neurogenin 3 did not identify JNK or FOXO1 as potential regulatory factors[15,18], suggesting that they may work upstream of neurogenin 3. This is supported by increased expression of *ATOH1*, a known secretory progenitor marker critical for neurogenin 3 expression[73,74], in RSP and AS enteroids (Supplementary Figs 2c and 3b). It is also possible that SP and AS are all having off target effects in our enteroid model. For example, SP600125 did not alter PDX1 expression in our enteroid system. Therefore, it is possible that JNK inhibition induces post-translational modifications on PDX1 or, that in human intestinal cells, JNK inhibition does not affect PDX1. Regardless of the mechanism, the use of RSP and AS induced differentiation of bona fide EE cells, as evidenced by our scRNA-seq data, which showed the presence of unique populations of cells that expressed EE markers and that were derived from populations of stem and progenitor cells.

Exposure of enteroids to combinations of Rim, SP, and AS led to the expression of specific hormones, with RSP inducing GIP expression and AS inducing higher levels of 5HT expression. Exposure to AS→RSP yielded similar levels of *GIP* mRNA and total GIP-positive enteroids, but increased levels of GIP secretion,

when compared to RSP alone; however, AS→RASP seemingly inhibited GIP protein production and secretion. AS alone appears to be the most potent inducer of 5HT secretion. Exposure to RSP, with or without AS, induced less 5HT secretion when compared to AS alone. Taken together, these combinatorial data suggest that AS exposure is a potent inducer of 5HT secretion while inhibiting GIP secretion, and exposure to Rim and SP is a potent inducer of GIP secretion while leading to reduced 5HT secretion when compared to AS treated enteroids. These data suggest that modulation of the endocannabinoid signaling system, JNK, and FOXO1, as well as other potential targets, could have multiple effects on EE function, controlling mRNA and protein production, as well as secretion, of multiple hormones.

In comparison to enteroids, rectoids showed a much stronger induction of EE cells when exposed to G2D12. It is clear that human small intestinal and colonic crypts demonstrate distinct gene expression profiles, which includes genes involved in epithelial cell differentiation[75], so it is likely that the requirements for EE cell lineage differentiation will be different between the various regions. Additional experiments examining whether the mechanisms defined for duodenal EE differentiation also regulate rectal EE differentiation need to be explored.

In summary, we have shown robust differentiation of human EE cells from duodenal and rectal ISCs using Wnt3a-containing differentiation media, and, with respect to enteroids, the addition of the small molecules rimonabant, SP600125, and AS1842856. These protocols improve upon current methods of EE cell differentiation without the use of direct genetic alteration. These studies also provide a platform for future experiments designed to identify endogenous factors regulating EE differentiation, identifying the role and response of EE cells in human GI disease, and further increasing EE cell numbers to potentially be used as personalized cell therapy.

## Methods

**Isolation of human intestinal crypts**. Tissues were procured as previously described[76]. In short, de-identified endoscopic and rectal biopsies were collected from grossly unaffected tissues in adolescent/young adult patients undergoing esophagogastroduodenoscopy and colonoscopy at Boston Children's Hospital for gastrointestinal complaints and used for organoid derivation. Further, de-identified duodenal resections were collected from adult patients undergoing pancreatico-duodenectomy at Massachusetts General Hospital prior to chemotherapy or radiation therapy for pancreatic carcinoma. Whole mucosal biopsies were generated from the duodenal resections using endoscopic biopsy forceps and used for total RNA isolation. Age and sex of donors can be found in Supplementary Table 2 but were unknown to researchers when experiments were being performed. Only macroscopically normal-appearing tissue was used from patients without a known gastrointestinal diagnosis. Each experiment was performed on at least three independent organoid lines derived from adolescent/young adult biopsy samples.

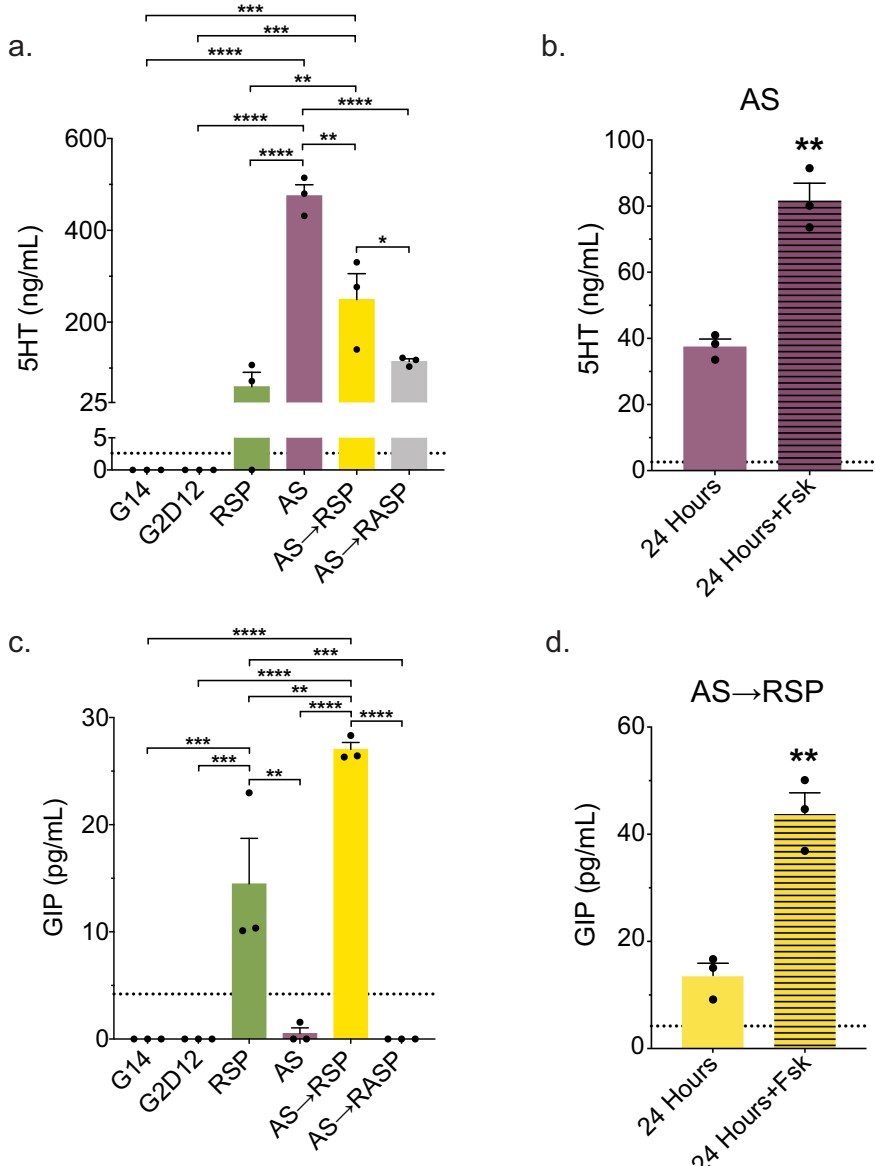

**Fig. 7 Differentiation Conditions Induce Hormone Secretion. a** Serotonin (5HT) ELISA of conditioned media from the last two days of differentiation of enteroids grown in G14, G2D12, RSP, AS, AS→RSP, and AS→RASP. Representative experiment showing $n = 3$ wells from each condition from a single enteroid line. *$p = 0.049$; **$p = 0.0049$ (RSP to AS→RSP), 0.0011 (AS to AS→RSP); ***$p = 0.0005$ (G14 to AS→RSP and G2D12 to AS→RSP); ****$p < 0.0001$. **b** 5HT ELISA of AS conditioned media collected after 24 h on day 13 (solid bar) and after 24 h with forskolin (Fsk) on day 14 (striped bar). Representative experiment showing $n = 3$ wells from each condition from a single enteroid line. **$p = 0.0015$. **c** Glucose-dependent insulinotropic peptide (GIP) ELISA of conditioned media from the last two days of differentiation of enteroids grown in G14, G2D12, RSP, AS, AS→RSP, and AS→RASP. Representative experiment showing $n = 3$ wells from each condition from a single enteroid line. **$p = 0.0013$ (RSP to AS), 0.0031 (RSP to AS→RSP); ***$p = 0.0009$ (G14 to RSP, G2D12 to RSP, and RSP to AS→RASP); ****$p < 0.0001$. **d** GIP ELISA of AS→RSP conditioned media collected after 24 h on day 13 (solid bar) and after 24 h with Fsk on day 14 (striped bar). Representative experiment showing $n = 3$ wells from each condition from a single enteroid line. **$p = 0.0025$. Bars show mean ± SEM; one-way ANOVA with Tukey correction for multiple comparisons (**a**, **c**); two-tailed unpaired $t$ test (**b**, **d**). Dotted line at 2.6 ng/mL represents the lower limit of detection for the 5HT ELISA (**a**, **b**). Dotted line at 4.2 pg/mL represents the lower limit of detection for the GIP ELISA (**c**, **d**). Each experiment repeated with at least three different enteroid lines. Source data are provided as a Source Data file.

Informed consent and developmentally appropriate assent were obtained at Boston Children's Hospital from the donors' guardian and the donor, respectively. Informed consent was obtained at Massachusetts General Hospital from the donors. All methods were approved and carried out in accordance with the Institutional Review Boards of Boston Children's Hospital (Protocol number IRB-P00000529) and Massachusetts General Hospital (Protocol number IRB-2003P001289).

Resections were briefly washed with pre-warmed DMEM/F12, after which the epithelial layer was separated from the rest of the duodenum manually with sterilized surgical tools then taken for RNA isolation. To isolate crypts, adolescent/young adult biopsies were digested in 2 mg/mL of Collagenase Type I reconstituted in Hank's Balanced Salt Solution for 40 min at 37 °C. Samples were then agitated by

pipetting followed by centrifugation at $500 \times g$ for 5 min at 4 °C. The supernatant was then removed, and crypts resuspended in 200–300 μL of Matrigel, with 50 μL being plated onto 4–6 wells of a 24-well plate and polymerized at 37 °C.

**Culturing of human duodenal and rectal organoids in vitro**. Isolated crypts in Matrigel were grown in specific growth media (GM) (Supplementary Table 1) based on the tissue of origin. The resulting organoids were passaged every 6–8 days as needed, with media changes occurring every two days. To passage, Matrigel was mechanically dissociated from the well and resuspended in 500 μL of Cell Recovery solution for 40–60 min at 4 °C. To aid in separating the Matrigel and enteroids, the tubes were gently inverted and then centrifuged at $500 \times g$ for 5 min at 4 °C. The

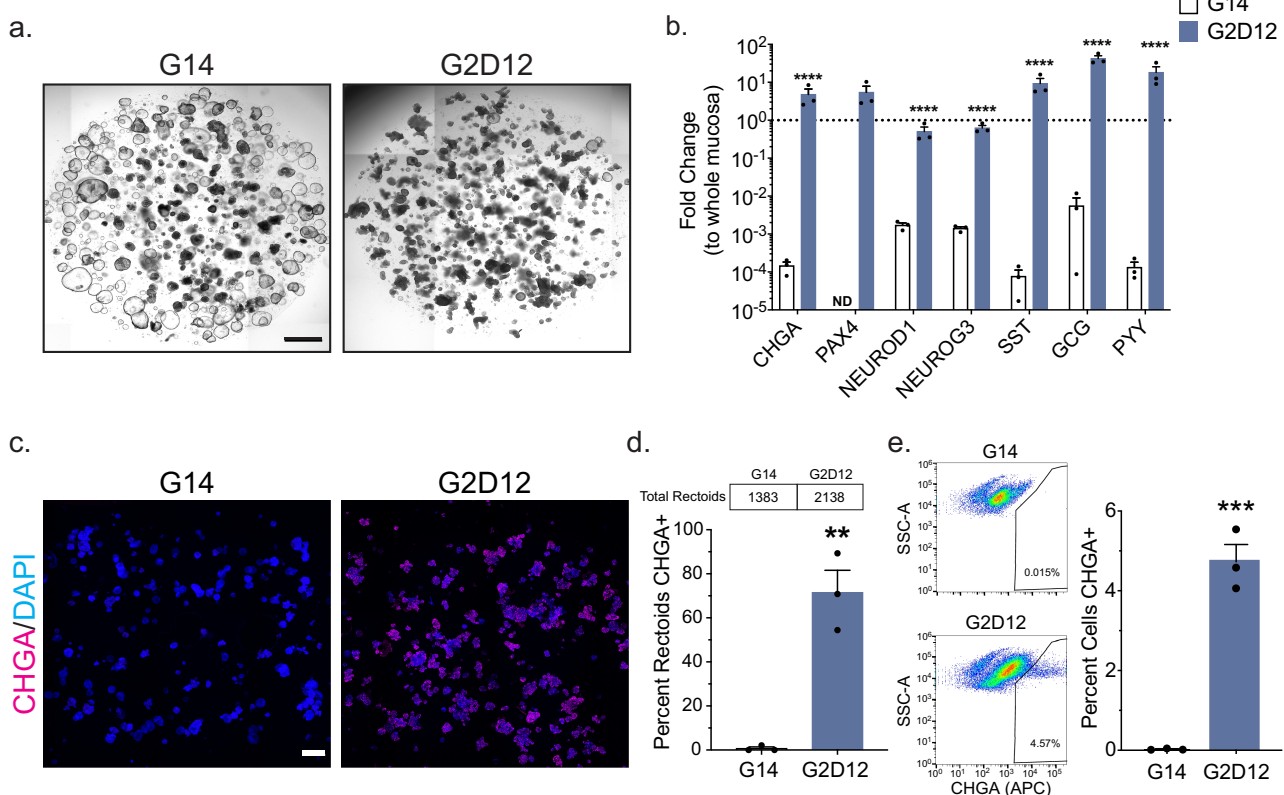

**Fig. 8 Induction of the Enteroendocrine Lineage in Rectoids. a** Representative light microscopy of rectoids (whole well) grown in G14 or G2D12. Scale bar = 1 mm. **b** qPCR analysis of EE markers of rectoids grown in either G14 or G2D12 compared to whole rectal mucosa (dotted line) and normalized to 18S. Representative experiment showing $n = 3$ wells for each condition from a single rectoid line. CHGA = chromogranin A, PAX4 = paired box 4, NEUROD1 = neuronal differentiation 1, NEUROG3 = neurogenin 3, SST = somatostatin, GCG = glucagon, PYY = peptide YY, ND = not detectable in one or more samples. ****$p < 0.0001$. **c** Representative immunofluorescence staining of CHGA (magenta) in rectoids (whole well) treated with G14 and G2D12. DNA (4′,6-diamidino-2-phenylindole (DAPI), blue). Scale bar = 1 mm. **d** Percentage of rectoids with CHGA staining in G14 and G2D12 treatments. Average results are from three different rectoid lines or passages. Table above graph shows the total number of rectoids examined per condition. **$p = 0.0022$. **e** Left two panels: Representative flow cytometry plots of CHGA + cells from rectoids grown in G14 and G2D12. Right panel: Percentage of CHGA + cells per well. Representative experiment showing $n = 3$ wells from each condition from a single rectoid line. ***$p = 0.0004$. Bars show mean ± SEM; two-way ANOVA with Tukey correction for multiple comparisons (**b**); two-tailed unpaired $t$ test (**d**, **e**). Each experiment repeated with at least three different rectoid lines. Specific conditions were excluded from statistical analysis if the data from one or more samples were labeled as not detectable. Source data are provided as a Source Data file.

supernatant was then removed, and organoids resuspended in Matrigel, followed by mechanical disruption via a bent-tipped pipette. Organoids were passaged at a 1:2 dilution, with 50 µL per well of a 24-well plate. After plating, the organoids were incubated at 37 °C for 10 min to allow the Matrigel to set. Once complete, 500 µL of GM was added to each well.

For differentiation, organoids were passaged and grown in GM for 2 days, to allow for ISC expansion, after which the organoids were transitioned to tissue-specific DM with additional small molecules added as described (Supplementary Table 1). Media was changed every two days, with Tubastatin A being removed after the second day of differentiation. Enteroid and rectoids were taken for analysis after 14 days.

**Gene expression analysis**. Total RNA was purified from individual wells using TRI®Reagent and the Direct-zol™ RNA kit, following the manufacturer's protocol. RNA concentration was determined using a NanoDrop™ 1000 spectrophotometer (Life Technologies). RNA was then reverse transcribed using the High-Capacity cDNA Reverse Transcription Kit. Gene expression analysis was then performed by Real-Time quantitative PCR (qPCR) using a QuantStudio 6 Flex thermocycler (Thermo Fisher). We used the following Taqman primers from Thermo Fisher (Supplementary Table 3). 18S transcripts were used as the internal control and data were expressed using the $2^{-ddCt}$ method with Ct limit of 40. Fold change, unless otherwise stated, was compared to total RNA derived from whole mucosal tissue biopsies, using duodenal mucosa with enteroids and rectal mucosa with rectoids.

**DNA isolation**. To isolate organoid genomic DNA, 200 µL of 50 mM NaOH was added to a single well of a 24-well plate and the Matrigel was mechanically dissociated. The samples were then transferred to 1.5 mL microcentrifuge tubes and placed in a 95 °C heat block for 20 min. The tubes were then vortexed, after which 25 µL of 1 M Tris-HCl was added. The samples were then centrifuged at 18,500 × $g$ for 10 min. The DNA content of the supernatant was then assayed using a NanoDrop™ 1000 spectrophotometer (Thermo Fisher).

**Immunofluorescence**. Immunofluorescence staining was performed as previously described with minor modifications[77]. Organoids were grown in and isolated from Matrigel as noted above. 1–3 wells from a 24-well plate were washed in 200 µL of 1x phosphate-buffered saline (PBS) and moved in suspension to a 1.5 mL microcentrifuge tube. Each tube was centrifuged at 800 × $g$ for 5 minutes at 4 °C to pellet organoids. PBS was aspirated, and organoids were fixed in 200 µL of 4% paraformaldehyde (PFA) for 20 min on ice, shaking. Each tube was centrifuged again as above, and PFA was aspirated. The organoids were then resuspended in 500 µL of 0.3% Triton-X in PBS and moved to a 48-well plate for the remaining steps. The organoids were permeabilized for 30 min at room temperature, shaking. Between each step, organoids were allowed to settle to the bottom of each well, the plate was angled, and the solution aspirated by careful pipetting. Organoids were then blocked with 5% bovine serum albumin (BSA) in PBS for 1.5 h at room temperature, shaking. This was followed by three five-minute washes in 500 µL of PBS at room temperature, shaking.

Each well was then incubated in 200 µL of primary antibodies diluted in 5% BSA/0.1% Trition-X in PBS at 4 °C overnight. This was followed by five washes in

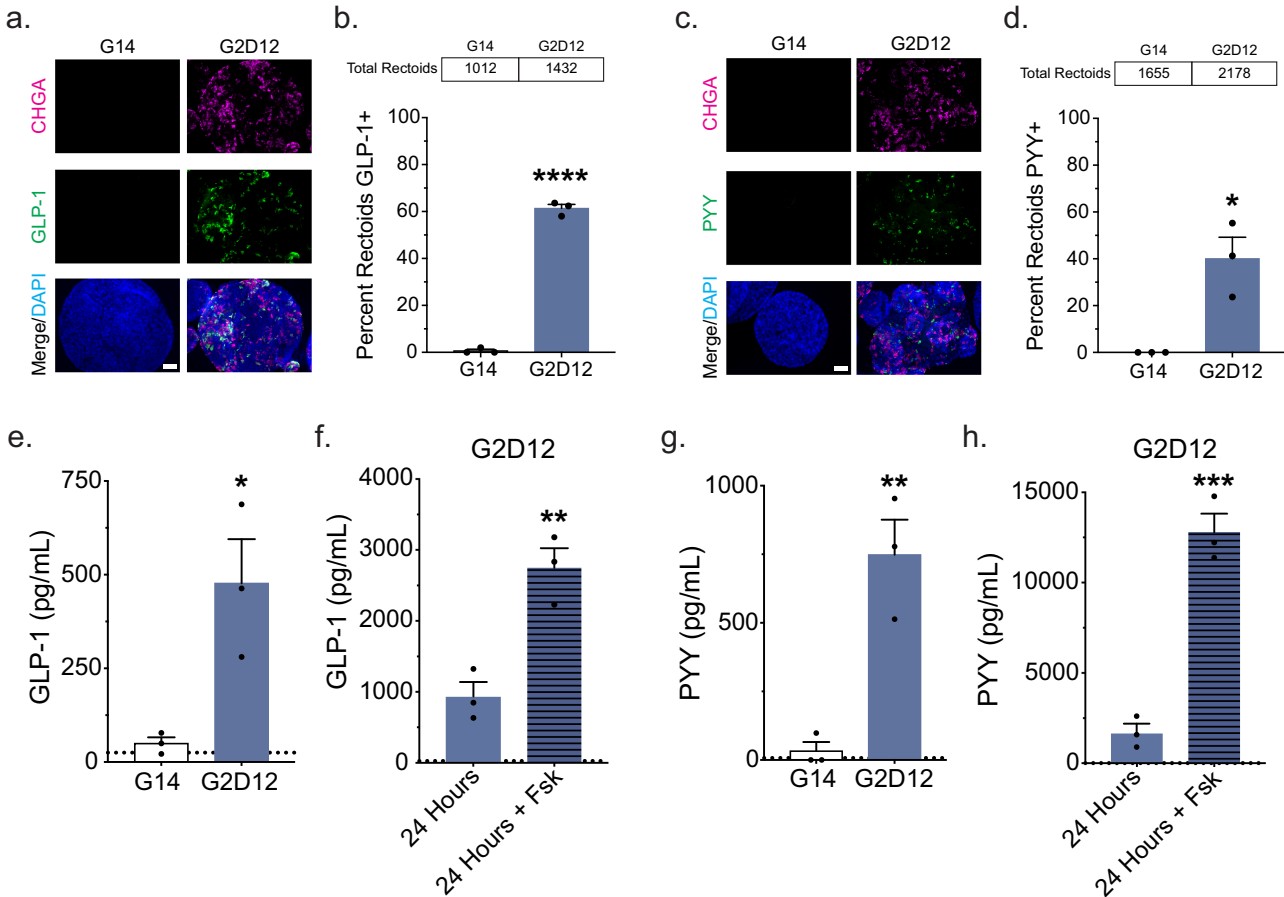

**Fig. 9 Hormone production and secretion in rectoids. a** Representative immunofluorescence staining of glucagon-like peptide-1 (GLP-1, green) and CHGA (magenta) in rectoids treated with G14 and G2D12. DNA (4′,6-diamidino-2-phenylindole (DAPI), blue). Scale bar = 50 μm. **b** Percentage of rectoids with GLP-1 staining in G14 and G2D12. Average results are from three different rectoid lines or passages. ****$p < 0.0001$. **c** Representative immunofluorescence staining of peptide YY (PYY, green) and CHGA (magenta) in rectoids treated with G14 and G2D12. DNA (DAPI, blue). Scale bar = 50 μm. **d** Percentage of rectoids with PYY staining in G14 and G2D12. Average results are from three different rectoid lines or passages. *$p = 0.0118$. **e** GLP-1 ELISA of conditioned media from the last two days of differentiation of rectoids grown in G14 and G2D12. Representative experiment showing $n = 3$ wells from each condition from a single rectoid line. *$p = 0.0229$. **f** GLP-1 ELISA of G2D12 conditioned media collected after 24 h on day 13 (solid bar) and after 24 hours with forskolin (Fsk) on day 14 (striped bar). Representative experiment showing $n = 3$ wells from each condition from a single rectoid line. **$p = 0.0061$. **g** PYY ELISA of conditioned media from the last two days of differentiation of rectoids grown in G14 and G2D12. Representative experiment showing $n = 3$ wells from each condition from a single rectoid line. **$p = 0.0056$. **h** PYY ELISA of G2D12 conditioned media collected after 24 h on day 13 (solid bar) and after 24 hours with forskolin (Fsk) on day 14 (striped bar). Representative experiment showing $n = 3$ wells from each condition from a single rectoid line. ***$p = 0.0006$. Bars show mean ± SEM; two-tailed unpaired $t$ test (**b**, **d**, **e**, **f**, **g**, **h**). Dotted line at 25 pg/mL represents the lower limit of detection for the GLP-1 ELISA (**e**, **f**). Dotted line at 7.3 pg/mL represents the lower limit of detection for the PYY ELISA (**g**, **h**). Each experiment repeated with at least three different rectoid lines. Tables above graphs show the total number of rectoids examined per condition. Source data are provided as a Source Data file.

500 μL of 0.1% Triton-X in PBS for 15 min each at room temperature, shaking. 200 μL of secondary antibodies diluted in 0.1% Triton-X in PBS were then added to each well and incubated for 2 h at room temperature, shaking. Organoids were then washed as above, then moved to new 1.5 mL centrifuge tubes. During the last wash, 4′,6-diamidino-2-phenylindole (DAPI) was added at a concentration of 1:1000 for nuclear staining. Organoids were then centrifuged at 1000 × g for 5 min at 4 °C to help remove as much PBS as possible. Slides were prepared by drawing three circles with a hydrophobic pen. Enteroids were then resuspended in 20 μL of Prolong Gold Antifade mountant and droplets placed within hydrophobic circles. The organoids were spread out to reduce clumping, sealed with a coverslip, and allowed to dry overnight at room temperature. Slides were stored at 4 °C for future imaging. Images were acquired using a Nikon upright Eclipse 90i microscope with a 20×/0.75 Plan-Apochromat objective and adjusted for brightness and contrast in Fiji (v2.10/1.53c)[78].

**Quantification of organoid immunofluorescence**. This technique was adapted from a previously described method[79]. Immunofluorescent images were acquired using an Invitrogen EVOS FL 2 Auto microscope (Thermo Fisher). Representative images of stained organoids were taken at 2× magnification. The stitched images were then processed in Fiji[78]. The color of the DAPI images was converted to 8-bit

grayscale and then the image was smoothed by applying a Gaussian Blur filter (radius = 4, scaled units). Thresholding of the smoothed images was performed using manual adjustment to achieve optimal separation of individual organoids. Watershed and Find Edges filters were then applied to segment any clumped organoids. Post-segmentation analysis was performed to outline and count individual organoids using Analyze Particles (size = 4000–100,000, circularity = 0.30–1.00, exclude on edges). Each image was then manually curated to exclude debris and organoids exhibiting background fluorescence. Any remaining clumped organoids were manually separated prior to quantification. The outlines generated from the DAPI images were then applied to the corresponding images from the other fluorescent channels. Each color image was converted to 8-bit grayscale and then the HiLo Lookup Table was applied. The threshold gate for stained cells was found by manual adjustment of positively stained organoids to achieve optimal representation. The threshold gate for each channel was then applied to each experimental condition. The Mean metric was extracted with ROI manager (measure) and compiled for analysis. Organoids with a Mean value of more than zero were considered positive.

**Flow cytometry**. Organoids were incubated in Cell Recovery solution for 40–60 min at 4 °C to remove the Matrigel and then centrifuged at 500 × g for 5 min at 4 °C. To achieve single-cell suspension, organoids were then incubated in 500 μL

of TrypLE Express at 37 °C for 30 min and broken up by repeated pipetting using a bent P1000 pipette tip. Each sample was then diluted in 800 µL of 20% fetal bovine serum (FBS) in Advanced DMEM/F12 and then centrifuged at 800 × g for 5 min at 4 °C. To mark dead cells, each sample was then incubated in DAPI (1:1000) diluted in 2% FBS/2 mM EDTA/calcium-free DMEM for 20 minutes at room temperature, then centrifuged at 800 × g for 5 min at 4 °C and washed in 2% FBS/2 mM EDTA/ DMEM. Cells were then incubated in 1% PFA for 15 min at room temperature, washed with 2% FBS/PBS and then permeabilized in 0.2% Tween 20 in 2% FBS/ PBS for 15 min at 37 °C. Following centrifugation, cells were resuspended in 0.1% Tween 20/2% FBS/2 mM EDTA in PBS with PE/Alexa Fluor 647-conjugated CHGA, PE-conjugated mouse IgG1, K isotype, Alexa Fluor 647-conjugated mouse IgG2b, K isotype, or with no antibody (the latter three acting as controls) for 30 min on ice. Cells were then washed in 0.1% Tween 20/2% FBS/2 mM EDTA in PBS, filtered through a 37-micron mesh, and then analyzed on a BD LSRFortessa flow cytometer using BD FACSDiva (v8.02) and FlowJo (v10.6.2). For gating strategy, please see Supplementary Fig 12.

**Cell proliferation**. Enteroid cell proliferation was assessed using the Click-IT EdU Alexa Fluor 488 Flow Cytometry Assay kit following the manufacturer's protocols. Briefly, enteroids were grown in Matrigel as noted above and taken for analysis after 7 and 14 days. Enteroids were labeled following incubation with 10 µM of EdU at 37 °C for 2 h and then incubated with TrypLE Express as above to isolate single cells. All samples were then diluted in 800 µL of 20% FBS in Advanced DMEM/F12 and then centrifuged at 800 × g for 5 min at 4 °C. Fixation, permeabilization, and the Click-iT reaction were performed in accordance with manufacturer's protocols before being filtered through a 37-micron mesh, and then analyzed on a BD LSRFortessa flow cytometer using BD FACSDiva and FlowJo.

**Cell apoptosis**. Apoptosis was assessed using the Dead Cell Apoptosis Kit with Annexin V Alexa Fluor 488 & Propidium Iodide following the manufacturer's protocols. For these experiments, enteroids were grown in Matrigel as noted above and taken for analysis after 7 and 14 days. To isolate single cells, enteroids were incubated with TrypLE Express as above. All samples were then diluted in 800 µL of 20% FBS in Advanced DMEM/F12 and then centrifuged at 800 × g for 5 min at 4 °C. Samples were then incubated with propidium iodide and annexin V per manufacturer's protocol. Cells were filtered through a 37-micron mesh, and then analyzed on a BD LSRFortessa flow cytometer using BD FACSDiva and FlowJo.

**ELISA**. Hormone quantification for 5HT, GIP, GLP-1, and PYY were performed using kits and following the manufacturer's protocols. For these experiments, organoids were grown in 48-well plates to aid in concentrating the hormone of interest. Conditioned media was taken on day 14 of differentiation, 48 hours after last media change. Diprotin A (100 µM), a dipeptidyl peptidase 4 inhibitor, was added daily to the media for the last 2 days of differentiation to prevent GIP and GLP-1 degradation. For hormone induction studies, conditioned media was taken on day 13, 24 h after last media change, after which forskolin (10 µM) was added to fresh media and collected after 24 h. As an additional control, DM not exposed to cells was also evaluated. This value was then subtracted from each experimental sample.

**Sample preparation for single cell RNA sequencing**. Enteroids were incubated in Cell Recovery solution for 40–60 min at 4 °C to remove the Matrigel and then centrifuged at 500 × g for 5 min at 4 °C. To achieve single cell suspension, organoids were then incubated in 500 µL of TrypLE Express at 37 °C for 30 min and broken up by repeated pipetting using a bent P1000 pipette tip. Each sample was then diluted in 800 µL of 20% FBS in Advanced DMEM/F12, filtered through 37 µm mesh, and then centrifuged at 300 × g for 10 min at 4 °C. Dead cells were then removed using MACS separation following the manufacturer's protocol. Briefly, each sample was resuspended in 100 µL Dead Cell Removal MicroBeads and incubated at room temperature for 15 min. Four hundred microliters of 1× Binding Buffer were then added to each sample, after which the samples were applied to prepared MACS columns placed in a MACS Separator. Flow-through containing live cells was collected. This was followed by washing the columns four times with 1× Binding Buffer, which were combined with the initial flow-through. These samples were then centrifuged at 800 × g for 5 min at 4 °C.

**Hashtag-labeling of sequencing samples**. Single cells isolated after dead cell removal were labeled prior to sequencing with Biolegend Totalseq-B hashtag antibodies, targeted against CD298 and β2-microglobulin, per the manufacturer's protocol[43]. Briefly, each of the nine samples (three separate differentiation conditions in three enteroid lines) were resuspended in 50 µL of Cell Staining Buffer and blocked with 5 µL of Human TruStain FcX™ blocking reagent for 10 min at 4 °C. The supernatant was then removed, and each sample resuspended in 50 µL of Cell Staining Buffer containing 1 µg of a unique Totalseq-B hashtag antibody. Samples were incubated for 30 min at 4 °C and then washed three times in 3.5 mL of Cell Staining Buffer. The samples were then filtered through a 37 µm mesh, resuspended in 2% BSA in PBS at a concentration of 1500 cells/µL, and then pooled together. Of note, prior to the scRNA-seq experiment, antibody binding was verified using fluorophore-conjugated antibodies with the same binding specificity.

**Droplet-based single-cell RNA sequencing and alignment**. A total of 180,000 cells as one pool were input across four channels, with a recovery rate of ~30,000 cells across all four channels. RNA libraries were prepared from single cells by the Single Cell Core at Harvard Medical School, Boston, MA, using the 10X Chromium Single Cell 3' Library Chip (v3.1) with dual indexed barcodes. Briefly, single cells were partitioned into Gel Beads in Emulsion by the Chromium Controller. Prior to partitioning, hashed cells were pooled together and "super-loaded" to reduce batch effects across samples[43]. Once partitioned, cells were lysed and RNA was captured, reverse transcribed, and amplified. cDNA was then enzymatically fragmented and sample indices were attached to both ends of the fragment. Following library prep, the transcriptomes were sequenced by the Molecular Biology Core Facilities at the Dana-Farber Cancer Institute on an Illumina NovaSeq 6000 sequencing platform. The pooled sample was split across two lanes on a NovaSeq S1 flow cell.

After sequencing, the files were processed, demultiplexed and aligned using 10X's Cell Ranger software (v5.0.1). Briefly, raw BCL files were demultiplexed using the cellranger mkfastq function, a wrapper of Illumina's bcl2fastq, with the filter dual index argument set to true. Following the generation of the FASTQ files, the samples were aligned to the hg19 human genome and features were counted using the cellranger count function. This function aligned the UMIs of each individual cell to the related gene and counted the number of detected UMIs to generate a feature-barcode matrix. In addition, it also measured the hashtag oligo counts for each individual cell from the relevant hashtag antibodies that were used to label each condition. Further, for analysis of RNA velocity, both un-spliced and spliced RNA variants were aligned and counted using the velocyto.py package (v0.17.16)[52].

**Single cell RNA sequencing analysis**. Analysis of the scRNA-seq data was conducted using the Seurat package (v4.0.1), within the R programming language (v4.0.3). Initially, cells were filtered to only include those with at least one detectable hashtag antibody signal, and then hashtag counts were normalized, and the cell samples were demultiplexed using the HTOdemux function built into Seurat. This function ranked a cell as being positive for a specific hashtag signal if the detected signal was in the 99th percentile or higher. Cells were then demultiplexed, with cells that had no positive signals being labeled as negative cells, cells with only one positive signal being labeled by their respective positive signal, and cells with two or more positive signals being labeled as doublets. Following demultiplexing, the dataset was pre-processed to remove cells that were deemed of lower quality, based on their number of unique genes (<200 unique genes) and the proportion of UMIs that corresponded to mitochondrial RNA (>25% of UMIs)[44].

After the removal of doublet barcodes, negative barcodes and the remaining barcodes that were deemed to be of lower quality, we were left with a combined dataset of 14,767 cells that was used for further analysis. To better control for batch effects and to improve our ability to detect diversity within our samples, we normalized the gene counts using regularized negative binomial regression, using SCTransform (v0.3.2)[80]. Following normalization, we performed dimensional reduction by running Principal Component Analysis and, based on the elbow plot generated, selected the first 20 principal components for defining the K nearest neighbors and the UMAP plot. After dimensional reduction, 15 clusters were defined using Louvain clustering (resolution = 0.5). Differentially expressed genes were identified across each cluster using the Wilcoxon rank sum test via the FindAllMarkers function built into Seurat. Only genes that were expressed in at least 25% of cells and had a log-fold change of 0.25 were considered in this analysis. These differential genes, as well as classical epithelial markers, were used to broadly categorize the clusters into six epithelial cell types and include *LGR5, AXIN2* and *ASCL2* for intestinal stem cells, *KRT20, FABP2*, and *FABP1* for enterocytes, *MUC2, FCGBP*, and *GFI1* for goblet cells, *CHGA, NEUROG3, NEUROD1*, and *PAX4* for enteroendocrine cells, *CD44* for progenitor cells and *MKI67* for proliferating cells[47–49].

EE cells were then subsetted out using Seurat's subset function and further analyzed. Similar to the parent dataset, gene counts were normalized using SCTransform and then dimensional reduction and cluster assignment was calculated using the Seurat pipeline. Across 471 cells, a total of 7 clusters were identified using Louvain clustering at a resolution of 0.8. Cluster-defining features were calculated using a Wilcoxon rank sum test through the FindAllMarkers function, with significant genes being defined as having been expressed in at least 5% of cells, with a log-fold change of 0.25 minimum. These genes, alongside markers previously used to annotate EE cell subsets, were used to assign identity to each cluster for downstream analysis. Using these markers, we broadly categorized the clusters into enteroendocrine progenitor cells (*DLL1, NEUROG3*), enterochromaffin cells (*CHGA, TPH1, NEUROD1*), and various peptide hormone producing subsets including M/X cells (*ARX, MLN, GHRL*) and G/K/I cells (*ARX, GAST, GIP, CCK*)[2,15,18,45,55,56]. Single cell gene expression of marker and hormone genes were visualized using Seurat's FeaturePlot function[81].

Finally, enteroid EE cell identity was compared to the gene set signature from tissue EE cells isolated from the murine small intestine[45]. To compare the gene signature, we calculated the gene module score of each individual cell using the AddModuleScore function found in Seurat v3 and v4, which calculates the score based on the enrichment of the specified gene set compared to randomly selected genes with a comparable average gene expression[44]. The AddModuleScore function calculates a scaled score between 0 and 1.

**Trajectory analysis with scVelo**. RNA velocity was calculated using the scVelo package (v0.2.3) with Scanpy (v1.7.1) on Python (v3.7.10)[51]. To perform trajectory analysis, the un-spliced and spliced variant count matrix that was previously calculated using velocyto (v0.17.15) was fused with an anndata object containing the UMAP information and cluster identities defined in Seurat analysis. The combined dataset was then processed using the scVelo pipeline: The ratio of un-spliced:spliced RNA for each gene was normalized and filtered using the default settings. Afterward, the first and second moments were calculated for velocity estimation. Following moment calculation, the dynamic model was used to calculate the RNA velocities. The dynamic model iteratively estimates the parameters that best model the phase trajectory of each gene, therefore capturing the most accurate, albeit more computationally intensive, estimate of the dynamics for each gene. These approaches were used to graphically model the RNA velocity for each culture condition.

**Quantification and statistical analysis**. All experiments were repeated using at least three different human organoid lines and representative data from a single line is shown, unless otherwise noted. For qPCR and flow cytometry studies, each condition was performed using pooled enteroids from 3-5 wells, unless otherwise noted, with each well acting as a single sample. Whole mucosal biopsies from duodenal resections and whole mucosa rectal biopsies were combined from three different individuals to generate a single reference sample for all enteroid and rectoid experiments.

Prior to statistical analysis, all qPCR data were transformed using $\log_{10}$. When analyzing only two conditions, statistical significance was determined by unpaired, two-tailed Student's $t$ test, combined with the Holm–Sidak method to control for multiple comparisons. When analyzing more than two conditions, statistical significance was determined by either one-way or two-way ANOVA, followed by Tukey post hoc analysis. Specific conditions were excluded from statistical analysis if the data from one or more samples were labeled as not detectable. To assess the proportion of cell populations within our scRNAseq data, we used tidyverse (1.3.1), PNWColors (0.1.0), and cowplot (1.1.1) to quantify and visualize our data. When analyzing the statistical relationship between cell proportions identified using single cell RNA sequencing, the nonparametric Kruskal-Wallis test was conducted, followed by Dunn's post-hoc analysis. We used the Bonferroni correction to account for multiple hypothesis testing when performing the post-hoc analysis. When comparing the distribution of gene module scores across the three different culture conditions, effect size was used to compare large changes and was calculated as Cohen's d. Statistical details for each experiment can be found within each figure legend.

**Reagents**. A list of all resources used in this study, including source and catalog number, can be found in Supplementary Table 4.

**Reporting summary**. Further information on research design is available in the Nature Research Reporting Summary linked to this article.

## Data availability
The scRNA-seq data generated in this study have been deposited in the Gene Expression Omnibus database under accession code GSE178342. All other data generated in this study are provided within the article and its Supplementary Information/Source Data file. Source data are provided with this paper.

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

## Acknowledgements

We thank members of the Breault laboratory, J. Majzoub, S. Bonner-Weir and J. Turner for helpful discussions. This research was supported by National Institute of Health (NIH) awards T32DK769937, K12DK09472109, the Juvenile Diabetes Research Foundation Advanced Postdoctoral Fellowship award 3-APF-2020-929-A-N, and Pediatric Endocrine Society Research Fellowship and Clinical Scholar Awards (to D.Z.), NIH awards HL095722 and UC4DK104165 (to J.M.K.), the HHMI Damon Runyon Cancer Research Foundation Fellowship DRG-2274-16, the AGA Research Foundation's AGA-Takeda Pharmaceuticals Research Scholar Award in IBD AGA2020-13-01, the HDDC Pilot and Feasibility P30 DK034854, and support from the Food Allergy Science Initiative, the Richard and Susan Smith Family Foundation and The New York Stem Cell Foundation as a Robertson Stem Cell Investigator (to J.O.M.), and NIH awards DK119488 and DK034854 and support from the Adolph Coors Foundation (to D.T.B.).

## Author contributions

D.Z. designed and performed experiments, analyzed data, prepared figures and co-wrote the manuscript; E.S., P.M., and W.Q. performed experiments, analyzed data and edited the manuscript; J.S.C. performed and analyzed the single cell RNA sequencing experiment; X.Y. and E.P.S. performed cell culture experiments; M.S.S. performed flow cytometry experiments; S.D. and S.H. assisted in qPCR and immunofluorescence experiments; J.M.K. provided conceptual advice and reagents; D.L.C. and J.O.M. provided conceptual advice and edited the manuscript, and D.T.B. directed the project and co-wrote the manuscript.

## Competing interests

The author J.M.K. holds equity in Frequency Therapeutics, a company that has an option to license IP generated by J.M.K. and that may benefit financially if the IP is licensed and further validated. Also, J.M.K. has been a paid consultant and/or equity holder for multiple companies (listed here https://www.karplab.net/team/jeff-karp). The interests of J.M.K. was reviewed and are subject to a management plan overseen by their institutions in accordance with their conflict-of-interest policies. J.O.M. reports compensation for consulting services with Cellarity and Hovione. The remaining authors declare no competing interests. A patent was filed by D.Z. and D.T.B covering compositions and methods for regulating enteroendocrine cell differentiation and uses thereof as described in the manuscript.

 
