## [Peer Review File · Nature Communications]

Reviewers' Comments:

Reviewer #1:

Remarks to the Author:

Robust Differentiation of human enteroendocrine cells from intestinal stem cells.

The manuscript by Zeve et al. describes a novel methodology for inducing the differentiation of some types of enteroendocrine cells from human adult tissue derived stem cells. The manuscript is easy to follow and well written. The methodology will be useful for those interested in exploring the function of human enteroendocrine cells, specifically GIP, SST and 5-HT producing cell types. However, I don't think the manuscript explores in enough depth the effects of the small molecules on the whole endocrine lineage or the mechanisms of action. There are several overstatements and a narrow view of the current literature. As such in its current form it is not suitable for publication in nature communications.

Specific comments.

1. A narrow view of the literature is presented; there are at least 5 papers describing human EEC differentiation using either small molecules or genetic induction of transcription factors and the induction of parts or the whole endocrine lineage. These are either ignored or not well referenced.

2. More detail is required in the introduction particularly with regard to the role of Gata4, JNK and Foxo1 roles in EEC differentiation, as it will make it easier for the reader to understand why the authors mainly investigating GIP and SST later in their result section

- Line 62. Chromogranin A as a marker of EE cells???(see Gehart et al). Chromogranin A is no longer considered a pan enteroendocrine marker. It marks progenitor Enterochromaffin cells which when fully differentiated express 5-HT. This view is supported by numerous sources using transgenic reporter mice and intestinal organoids from both mouse and humans.
- Line 66: I would not categorise Pdx1 as a critical TF for EE cell differentiation as it is only shown to have a role in proximal small intestine. I had to read the provided ref to find that inactivation of Pdx1 in duodenum leads to reduce mRNA levels of GIP and SST. Therefore Pdx 1 is important for the differentiation of a small proportion of EEC subtypes.
- Since the intestinal segment in study is the duodenum and it produces most of the CCK, why is it not quantified? Ref 21 was used as basis in this manuscript for GATA4 role in EEC identity and specifically in GIP-expressing cells. Said reference also mentions a decrease in Cck after Gata4-Gata6 double conditional knockout, although the expression of this transcript was not quantified
- Line 77: A bit misleading as it refers to the role of JNK signalling in endocrine cells (references 23 and 24 review papers, showing the effect in pancreatic beta cells). Any info in literature of the role of JNK signalling in Pdx1 regulation in the intestine?
- I understand that since the authors are interested in GATA4, JNK and FOXO1 they only show data on SST and GIP. But Fig1 shows their DM before addition of any modulators. What about expression other EECs markers like CCK or 5-HT, in G14 and G2D12 medium?
- What about the expression of other TFs? Like Pax4?
- Together the above points all show the described methodology is specific for a limited range of gut hormones, and yet the authors consistently describe their effects as if they influence all EEC's, this is misleading. It will be important to explore the effects on the full endocrine lineage. This would be best completed using single cell RNAseq or similarly unbiased method.
- Have they study the effects of their differentiation protocol (the one before adding modulators of GATA4, JNK and Foxa1) in other parts of the intestine?
- Line 167-169: Moreover, compared to enteroids grown in G14 and G2D12, treatment with RSP led to the upregulation of multiple EE markers (CHGA, PDX1, NEUROD1, NEUROG3, SST, and GIP). Not correct statement for Pdx1 when compared to G2D12
- Suppl Fig 2: can the authors explain the increase in Lgr5 in RSP and AS medium?
- Equally the effect of the small molecules alone or in combination on growth and survival has not been well explored. The authors should consider more detailed methods for tracking organoid growth and apoptosis. This has important implications for differentiation.
- Authors suggest that these molecules work upstream of Ngn3. Why not quantify Math1?
- No attempt has been made to correlate the differentiated cells identity with that of native EEC identity.
- The secretion studies do not measure secretion rather hormone leak. EEC's respond rapidly to stimuli to release their hormones in a similar fashion to beta-cells in the islet or neuronal cells. To suggest that hormone release can be quantified after 48 hours when the normal response would be in seconds is nonsense. Secretion of gut hormones should be stimulated with a known activator or at

the very least IBMX and forskolin and measured over 1-2hours. Without this data there is no way of telling if the cells produced by small molecule protocol are functional.

- It is of paramount importance the authors demonstrate the identity of the cells they are differentiating and how the small molecules affect EEC fate decisions. I suspect the model does not upregulate the differentiation of the whole endocrine lineage and is likely to be specific to a subset of cells. Do these cells represent a true native EEC phenotype or an upregulation of specific gut hormone transcripts within cells that would not otherwise express them? Without this knowledge or the correct evidence demonstrating functionality the model has poor value as a tool for understanding EEC's.

Reviewer #2:

Remarks to the Author:

The goal of this study by Zeve et al. is to develop a small molecule-based cell culture protocol to enhance the presence and function of enteroendocrine (EE) cells in a human enteroids. The paper synthesizes previous knowledge of enteroendocrine cell development and differentiation to propose a hypothesis that modulation of GATA4, PDX1, and FOXO1 activities will enhance EE cell numbers in enteroid culture. The paper will be of interest to those working in the field of GI biology and disease because it provides a methodology to enhance EE cell numbers in duodenal enteroid cultures. Given that EEs are a difficult cell to study because i) they are a rare intestinal cell type with multiple sub-types making any specific EE cell type quite rare; and ii) there are not adequate cell culture models for these cells. The work does not particularly influence thinking in the field because it is a techniques study more than a mechanistic study. However, that is not to mean that the novel culture scheme presented to enhance EE culture in human enteroids is not valuable. The data are convincing. Experiments are well designed and well controlled. The manuscript is logically presented and well written. The figures are well presented. One limitation of the work is that the applicability of the protocol beyond the duodenum is unclear and not tested. Therefore, a key experiment that would strengthen the paper would be to apply the protocol to enteroids from other regions of the GI tract to determine how universally applicable the protocol is. Thinking about the bigger biological question, experiments to delineate/test mechanisms specific mechanisms at play in terms of how the small molecules actually work on the targeted TFs to enhance EE cells in duodenal enteroids would be outstanding. But, it is understandable that such experiments may reach beyond the goal of this study, which is primarily to report a valuable technical advance. Additional specific comments are presented below.

No data are presented to validate that the small molecules used modulate the function of the TFs targeted (GATA4, PDX1, FOXO1).

It was surprising that CCK wasn't examined given that it is an abundant proximal intestine EE cell type, and it has been shown to be affected by changes in GATA proteins.

There is some confusion about the rationale/proposal that rimonabant works via GATA4 to enhance EE cells in culture. GATA4 is not expressed in EEs. The data referenced for studies of the GATA4-GIP relationship (Jepeal et al., 2008) are not strong. The staining in that paper showing co-expression of GATA4 and GIP in mouse duodenum is questionable, and the studies done to show GATA4 modulates GIP expression were done with a neuroendocrine tumor cell line subclone. If GATA4 is the target, it, therefore, likely acts through a non-cell autonomous pathway. This is not discussed in the manuscript. Of course, whether or not GATA4 acts cell autonomously or non-cell autonomously does not call into question the data demonstrating that rimonabant enhances EE culture; it just raises questions about how the small molecule works. If not examined experimentally, it should at least be discussed. On a similar thread, GATA6, unlike GATA4, is expressed in EEs. GATA6 has also been shown to affect EEs in mouse models. There seems to be at least a formal possibility that rimonabant could increase GATA6 activity in enteroids to enhance EE cells given the similarities of these GATA factors in terms of function.

A minor question relates to normalization of gene expression to mucosal levels. It is elegant to compare gene expression levels in enteroids with mucosal levels. It would be helpful for the authors to provide the rationale for comparing levels in pediatric enteroids (age range 13-21 years) with those in adult mucosa (age range 55-82 yrs).

Reviewer #3:

Remarks to the Author:

Enteroendocrine cells (EECs) are the largest population of endocrine cells in humans and are essential regulators of many homeostatic processes and functions. However, there are significant challenges in deriving EECs from human intestinal stem cells (ISCs) in-vitro that prevent investigations into their role in disease and homeostasis. Current approaches use inhibition of Wnt, Notch, MAPK, and/or BMP signaling to induce EEC differentiation, to limited success. Zeve et. al. employed a different approach to address these limitations in current protocols. By using small molecule targeting of specific transcriptional regulators, the authors were able to massively increase differentiation of hEECs from hISCs in organoids as measured by immunostaining, qPCR, and FACS for specific markers of the EEC lineage. Interestingly, they also showed the ability to tailor the make-up of the induced EECs by changes in expression levels for CHGA, 5HT, SST, and GIP, important secretions produced by endocrine cells, by changing the duration and sequence of the three inhibitors that they studied. The authors concluded that their approach significantly improved over current protocols for induction of the EEC lineages in hISCs. Moreover, their results also contradict much evidence in current literature that purport that inhibition of Wnt3A production is required to induce EECs. Instead, Zeve et. al. find that removal of Wnt3A is detrimental to induction efforts and long-term viability of cultured organoids. The authors data provide clear evidence for their approach as a more successful approach in producing EECs as well as their RNA and protein markers in in vitro organoid culture than conventional methods. The approach that the authors use is novel, well-thought through, and based on deep understanding of the complexity involved in regulation of hISC differentiation. Their work represents a building block for others working in this field to build upon and further advance the understanding of the role of EECs in the human intestine.

1. Minor – The authors briefly characterize other lineages in the enteroids under their conditions through qPCR under the different DM conditions (in supplemental data). It would be important to see IF for some other markers of intestinal lineages to see how their DM conditions affected the expression of other lineage markers at the protein level.
2. Minor – In Figure 2, specify time point of photo for RSP enteroids
3. Authors need to list all vendors or catalog numbers for many of the components that they use in their medium recipes such that others can reproduce their results.

RESPONSE TO REVIEWER COMMENTS:

We thank the reviewers for their thoughtful and insightful comments which have allowed us the opportunity to significantly bolster our findings and conclusions. We have now expanded our analysis to include expression of cholecystokinin (CCK), a critical duodenal hormone, as well as analysis of the proliferation and apoptosis of enteroid cells during our differentiation protocols. In response to reviewer comments, we have also performed single cell sequencing analysis, which has provided important new insights regarding the dynamics of enteroendocrine cell differentiation. In addition, we have extended our studies to include the distal GI tract (rectum), which, combined with our studies of duodenal enteroids, demonstrate regional specificity in the production of enteroendocrine hormones. Finally, to provide additional functional validation regarding the EE cells we have generated, we demonstrate that secretion of select hormones is highly responsive to stimulation with forskolin. All rebuttal comments, and major changes within the manuscript, are denoted with blue text.

Reviewer #1 (Remarks to the Author):

Robust Differentiation of human enteroendocrine cells from intestinal stem cells.

The manuscript by Zeve et al. describes a novel methodology for inducing the differentiation of some types of enteroendocrine cells from human adult tissue derived stem cells. The manuscript is easy to follow and well written. The methodology will be useful for those interested in exploring the function of human enteroendocrine cells, specifically GIP, SST and 5-ht producing cell types. However, I don't think the manuscript explores in enough depth the effects of the small molecules on the whole endocrine lineage or the mechanisms of action. There are several overstatements and a narrow view of the current literature. As such in its current form it is not suitable for publication in nature communications.

Specific comments.

1. A narrow view of the literature is presented; there are at least 5 papers describing human EEC differentiation using either small molecules or genetic induction of transcription factors and the induction of parts or the whole endocrine lineage. These are either ignored or not well referenced.

We apologize for the omission of these important references. We have now included additional references (14, 18-21) in the Introduction regarding the use of genetic induction of NEUROG3 in the study of enteroendocrine differentiation. In addition, we have included additional references (8,9,16,22,23) highlighting studies that employ specific small molecules to induce enteroendocrine differentiation. (Please see changes on page 3-4, lines 73-78)

2. More detail is required in the introduction particularly with regard to the role of Gata4, JNK and Foxo1 roles in EEC differentiation, as it will make it easier for the reader to understand why the authors mainly investigating GIP and SST later in their result section

We thank the reviewer for this helpful suggestion. We have now added more details to explain the reasoning behind targeting GATA4, JNK and FOXO1 in the Introduction. For example, we now highlight the previously known role of GATA4 in GIP and CCK expression, the role of JNK in PDX1 and CHGA expression, and the role of FOXO1 in enteroendocrine development. (Please see changes on page 4 lines 80-99)

• **Line 62. Chromogranin A as a marker of EE cells???(see Gehart et al). Chromogranin A is no longer considered a pan enteroendocrine marker. It marks progenitor Enterochromaffin cells which when fully differentiated express 5-ht. This view is supported by numerous sources using transgenic reporter mice and intestinal organoids from both mouse and humans.**

Thank you for pointing this out. We have now updated this statement to note that EE cells are defined by the specific hormones they produce and that they can express multiple neuroendocrine secretory proteins, including CHGA. (Please see changes on page 3, lines 64-67)

• **Line 66: I would not categorise Pdx1 as a critical TF for EE cell differentiation as it is only shown to have a role in proximal small intestine. I had to read the provided ref to find that inactivation of Pdx1 in duodenum leads to reduce mRNA levels of Gip and Sst. Therefore Pdx 1 is important for the differentiation of a small proportion of EEC subtypes.**

We agree with the reviewer's comment and have updated the Introduction to reflect that PDX1 regulates EE gene expression within the duodenum. (Please see changes on page 3, lines 69-70)

• **Since the intestinal segment in study is the duodenum and it produces most of the CCK, why is it not quantified? Ref 21 was used as basis in this manuscript for GATA4 role in EEC identity and specifically in GIP-expressing cells. Said reference also mentions a decrease in Cck after Gata4-Gata6 double conditional knockout, although the expression of this transcript was not quantified**

Thank you for this insightful comment. We have now expanded our analysis to include gene and protein expression of CCK. Interestingly, treatment with RSP leads to higher CCK expression compared to treatment with AS, suggesting that RSP could be working, at least in part, through activation of GATA4. (Please see Figures 1b, 2b, 3b, 5a, 6g, and 6h)

• **Line 77: A bit misleading as it refers to the role of JNK signalling in endocrine cells (references 23 and 24 review papers, showing the effect in pancreatic beta cells). Any info in literature of the role of JNK signalling in Pdx1 regulation in the intestine?**

Thank you for this important point. We have now updated the statement to directly discuss the role of JNK signaling in beta cells. In addition, we have included an additional report suggesting JNK2 may have a role in regulating murine EE cell differentiation (33). Finally, we emphasize that the role of JNK signaling in the regulation of PDX1 has not yet been evaluated during the directed differentiation of human ISCs. (Please see changes on page 4, lines 84-89)

• **I understand that since the authors are interested in GATA4, JNK and FOXO1 they only show data on SST and GIP. But Fig1 shows their DM before addition of any modulators. What about expression other EECs markers like CCK or 5-HT, in G14 and G2D12 medium?**

We have now included expression and secretion data for multiple duodenal hormones, including CCK and 5HT, in response to all conditions, including G14 and G2D12. (Please see Fig 1b, 2b, and 3b for gene expression and Figures 6 and 7 and Supplementary Fig 8 for protein expression and secretion)

• **What about the expression of other TFs? Like Pax4?**

We apologize for the omission of other transcription factors from our original analysis. We have now included gene expression of PAX4 and ATOH1 throughout the manuscript. (Please see Figures 1b, 2b, 3b, 5a, 8b and Supplementary Figures 1a, 2c, 3b, 7a, 9a)

• **Together the above points all show the described methodology is specific for a limited range of gut hormones, and yet the authors consistently describe their effects as if they influence all EEC's, this is misleading. It will be important to explore the effects on the full endocrine lineage. This would be best completed using single cell RNAseq or similarly unbiased method.**

Thank you for this insightful point. We have now performed a single cell RNA sequencing analysis comparing enteroids treated with G2D12, AS or RSP and have identified a distinct cluster that represents enteroendocrine cells. This group of cells expresses a majority of known duodenal enteroendocrine cell gene markers. Moreover, unbiased RNA velocity analysis suggests these enteroendocrine cells are derived from multipotent intestinal progenitor cells. (Please see pages 9-13, lines 220-315, Figure 4, and Supplementary Figure 4)

- **Have they study the effects of their differentiation protocol (the one before adding modulators of GATA4, JNK and Foxa1) in other parts of the intestine?**

We thank the reviewer for this important suggestion. To explore this, we obtained rectal biopsies to study the impact of our G2D12 differentiation protocol in the distal GI tract. Unlike in duodenal enteroids, G2D12 conditions proved to be sufficient for induction of EE cell markers in human rectoids, including the expression and secretion of GLP-1 and PYY. These findings suggest that intestinal stem cells from different parts of the intestine have different requirements for EE cell differentiation. (Please see page 17, lines 414-433, Figure 8, and Supplementary Figure 9)

- **Line 167-169: Moreover, compared to enteroids grown in G14 and G2D12, treatment with RSP led to the upregulation of multiple EE markers (CHGA, PDX1, NEUROD1, NEUROG3, SST, and GIP). Not correct statement for Pdx1 when compared to G2D12**

Thank you for noticing this. We have removed PDX1 from that statement. (Please see changes on page 8, lines 190-192)

- **Suppl Fig 2: can the authors explain the increase in Lgr5 in RSP and AS medium?**

We thank the reviewer for this very interesting question. While we don't have a formal explanation, it is tempting to speculate. For instance, a prior study has suggested that EE cells can act as facultative stem cells, being activated in response to stress or injury (61). Therefore, it may be that as the enteroids age in our culture system, or possibly due to the loss of Matrigel integrity, some of the EE cells may function as facultative stem cells. This point has now been added to the Discussion, but we respectfully submit that further investigation of this question is beyond the scope of this manuscript. (Please see changes on page 20, lines 488-492)

- **Equally the effect of the small molecules alone or in combination on growth and survival has not been well explored. The authors should consider more detailed methods for tracking organoid growth and apoptosis. This has important implications for differentiation.**

This is a very salient point that has now been addressed in the manuscript. To assess proliferation and apoptosis, we performed short-term EdU uptake and annexin V labeling studies, respectively, on Days 7 and 14. We found that on Day 7 the enteroids exposed to growth media had a much higher rate of proliferation compared to enteroids exposed to the differentiation conditions. By Day 14, however, no difference in proliferation was observed between groups.

Enteroids exposed to growth media also showed less apoptosis compared to enteroids exposed to the differentiation conditions on Day 7; however, by Day 14, AS exposed enteroids showed significantly lower levels of annexin V labeling compared to G14, while there was no significant difference in Annexin V labelling between G2D12 and AS. (Please see pages 14-15, lines 359-371 and Supplementary Figures 7e and 7f)

- **Authors suggest that these molecules work upstream of Ngn3. Why not quantify Math1?**

We thank the reviewer for this suggestion. We have now included analysis of ATOH1 gene expression throughout the manuscript. Of note, its expression is increased in response to both AS and RSP when compared to G14 and G2D12 treated enteroids and increased in G2D12 rectoids. (Please see Supplementary 1a, 2c, 3b, 7a, 9a)

• No attempt has been made to correlate the differentiated cells identity with that of native EEC identity.

We thank the reviewer for this very important point. To address this, we have taken two independent approaches. First, as a reference population, we employed mRNA from human mucosal biopsies, from duodenum and rectum, in our qPCR experiments. This allows us to compare gene expression levels in our organoids to native tissue levels for a broad array of cell types, including EE cells. Second, our single cell RNA sequencing analysis allowed us to compare a known lists of EE cell markers to those identified in response to our induction conditions, which showed striking similarities. The most thorough single RNA sequencing experiment would involve a direct comparison of cells isolated directly from duodenal biopsies to those induced by our protocols, which is beyond the scope of this study.

• The secretion studies do not measure secretion rather hormone leak. EEC's respond rapidly to stimuli to release their hormones in a similar fashion to beta-cells in the islet or neuronal cells. To suggest that hormone release can quantified after 48 hours when the normal response would be in seconds is nonsense. Secretion of gut hormones should be stimulated with a known activator or at the very least IBMX and forskolin and measured over 1-2hours. Without this data there is no way of telling if the cells produced by small molecule protocol are functional.

To address this important point, we have performed additional experiments using forskolin to induce hormone secretion and analyzed additional time points. Unfortunately, the 1–2-hour timepoint had high variability. Therefore we utilized a protocol established in Hans Clevers' laboratory (18) to assess hormone secretion in response to forskolin at 24-hours, which revealed strong induction of hormone secretion for 5HT, GIP, PYY and GLP-1 when compared to organoids not exposed to forskolin. (Please see Figures 7b, 7d, 8k, and 8m and Supplementary Figures 8b, 8d, 9c, and 9e)

• It is of paramount importance the authors demonstrate the identity of the cells they are differentiating and how the small molecules affect EEC fate decisions. I suspect the model does not upregulate the differentiation of the whole endocrine lineage and is likely to be specific to a subset of cells. Do these cells represent a true native EEC phenotype or an upregulation of specific gut hormone transcripts within cells that would not otherwise express them? Without this knowledge or the correct evidence demonstrating functionality the model has poor value as a tool for understanding EEC's.

We appreciate the reviewer raising this important point. Using our single cell RNA sequencing analysis, we show that the EE cells induced by our protocols represent a true duodenal EE cell population, seemingly being derived from intestinal progenitor cells. By demonstrating the expression of early, intermediate and late markers of the enteroendocrine lineage, in addition to demonstrating functionality through forskolin-induced secretion of gut hormones, we believe that our model will be of great value to the scientific community.

Reviewer #2 (Remarks to the Author):

The goal of this study by Zeve et al. is to develop a small molecule-based cell culture protocol to enhance the presence and function of enteroendocrine (EE) cells in a human enteroids. The paper synthesizes previous knowledge of enteroendocrine cell development and differentiation to propose a hypothesis that modulation of GATA4, PDX1, and FOXO1 activities will enhance EE cell numbers in enteroid culture. The paper will be of interest to those working in the field of GI biology and disease because it provides a methodology to enhance EE cell numbers in duodenal enteroid cultures. Given that EEs are a difficult cell to study because i) they are a rare intestinal cell type with multiple sub-types making any specific EE cell type quite rare; and ii) there are not adequate cell culture models for these cells. The work does not particularly influence thinking in the field because it is a techniques study more than a mechanistic study. However, that is not to mean that the novel culture scheme presented to enhance EE culture in human enteroids is not valuable. The data are convincing. Experiments are well designed and well controlled. The manuscript is logically presented and well written. The figures are well presented. One limitation of the work is that the applicability of the protocol beyond the duodenum is unclear and not tested. Therefore, a key experiment that would strengthen the paper would be to apply the protocol to enteroids from other regions of the GI tract to determine how universally applicable the protocol is. Thinking about the bigger biological question, experiments to delineate/test mechanisms specific mechanisms at play in terms of how the small molecules actually work on the targeted TFs to enhance EE cells in duodenal enteroids would be outstanding. But, it is understandable that such experiments may reach beyond the goal of this study, which is primarily to report a valuable technical advance. Additional specific comments are presented below.

We thank the reviewer for these supportive and constructive comments. As highlighted in response to Reviewer 1, to expand the applicability of our model, we have extended our studies to include human rectoids. Remarkably, exposure to our G2D12 conditions proved to be sufficient for induction of EE cell markers in these cells, including the expression and secretion of GLP-1 and PYY. These findings suggest that ISCs from different regions of the intestine have different requirements for EE cell differentiation. (Please see page 17, lines 414-433, Figure 8, and Supplementary Figure 9)

No data are presented to validate that the small molecules used modulate the function of the TFs targeted (GATA4, PDX1, FOXO1).

Thank you for bringing up this very important point. SP600125 and AS1842856 are both very well-described small molecule inhibitors for their respective targets, JNK and FOXO1. For example, SP600125 has been used in over 30,000 studies to inhibit JNK while AS1842856 has been used in over 400 studies to inhibit FOXO1 (data per Google Scholar). The function of rimonabant as a GATA4 activator, on the other hand, is much less studied. We have shown indirectly that rimonabant may function through GATA4 by showing increased expression of both CCK and GIP, known targets of GATA4, when enteroids are exposed to rimonabant; however, it is quite possible that rimonabant is not functioning through activation of GATA4 and additional validation is necessary but is beyond the scope of this paper. We have now included a much more thorough discussion regarding the possible mechanisms through which rimonabant works. (Please see changes on page 20, lines 499-512)

It was surprising that CCK wasn't examined given that it is an abundant proximal intestine EE cell type, and it has been shown to be affected by changes in GATA proteins.

We thank the reviewer for the suggestion to include CCK to our analysis. As pointed out in response to Reviewer 1, we have now expanded our analysis to include gene and protein expression of CCK. Interestingly, RSP treatment leads to higher CCK expression compared to AS, suggesting that RSP, at least in part, could be working through activation of GATA4. (Please see Figures 1b, 2b, 3b, 5a, 6g, and 6h)

There is some confusion about the rationale/proposal that rimonabant works via GATA4 to enhance EE cells in culture. GATA4 is not expressed in EEs. The data referenced for studies of the GATA4-GIP relationship (Jepeal et al., 2008) are not strong. The staining in that paper showing co-expression of GATA4 and GIP in mouse duodenum is questionable, and the studies done to show GATA4 modulates GIP expression were done with a neuroendocrine tumor cell line subclone. If GATA4 is the target, it, therefore, likely acts through a non-cell autonomous pathway. This is not discussed in the manuscript. Of course, whether or not GATA4 acts cell autonomously or non-cell autonomously does not call into question the data demonstrating that rimonabant enhances EE culture; it just raises questions about how the small molecule works. If not examined experimentally, it should at least be discussed. On a similar thread, GATA6, unlike GATA4, is expressed in EEs. GATA6 has also been shown to affect EEs

in mouse models. There seems to be at least a formal possibility that rimonabant could increase GATA6 activity in enteroids to enhance EE cells given the similarities of these GATA factors in terms of function.

Thank you for making this very important point. We agree that it is unclear whether rimonabant is functioning through activation of GATA4 and/or through other mechanisms. We have increased our discussion of this, as well as included some discussion of GATA6. (Please see changes on page 20, lines 499-512)

A minor question relates to normalization of gene expression to mucosal levels. It is elegant to compare gene expression levels in enteroids with mucosal levels. It would be helpful for the authors to provide the rationale for comparing levels in pediatric enteroids (age range 13-21 years) with those in adult mucosa (age range 55-82 yrs).

We thank the reviewer for this question regarding the choice of adolescent/young adult versus adult samples. Due to the availability of abundant duodenal tissue from adult surgical resections, we chose this source to normalize gene expression to mucosal levels. In contrast, adolescent/young adult samples were only available as endoscopic biopsies, a precious and limiting resource, which were consistently used to generate enteroids. Since submission of the manuscript, we have compared whole mucosal gene expression from adult and adolescent/young adult samples (three of each). Despite some variability in expression between samples, we did not identify any significant differences between the two groups. The data are presented as fold change to adult samples (Mean \pm SEM): **CHGA** – Adult (1.020 \pm 0.144), Adol (5.075 \pm 3.101); **MUC2** – Adult (1.179 \pm 0.392), Adol (0.690 \pm 0.301); **LYZ** – Adult (1.222 \pm 0.567), Adol (23.423 \pm 18.427); **ALPI** – (1.199 \pm 0.510), Adol (4.339 \pm 1.139) and **LGR5** – Adult (1.001 \pm 0.036), Adol (7.888 \pm 5.972).

Reviewer #3 (Remarks to the Author):

Enteroendocrine cells (EECs) are the largest population of endocrine cells in humans and are essential regulators of many homeostatic processes and functions. However, there are significant challenges in deriving EECs from human intestinal stem cells (ISCs) in-vitro that prevent investigations into their role in disease and homeostasis. Current approaches use inhibition of Wnt, Notch, MAPK, and/or BMP signaling to induce EEC differentiation, to limited success. Zeve et. al. employed a different approach to address these limitations in current protocols. By using small molecule targeting of specific transcriptional regulators, the authors were able to massively increase differentiation of hEECs from hISCs in organoids as measured by immunostaining, qPCR, and FACS for specific markers of the EEC lineage. Interestingly, they also showed the ability to tailor the make-up of the induced EECs by changes in expression levels for CHGA, 5HT, SST, and GIP, important secretions produced by endocrine cells, by changing the duration and sequence of the three inhibitors that they studied. The authors concluded that their approach significantly improved over current protocols for induction of the EEC lineages in hISCs. Moreover, their results also contradict much evidence in current literature that purport that inhibition of Wnt3A production is required to induce EECs. Instead, Zeve et. al. find that removal of Wnt3A is detrimental to induction efforts and long-term viability of cultured organoids. The authors data provide clear evidence for their approach as a more successful approach in producing EECs as well as their RNA and protein markers in in vitro organoid culture than conventional methods. The approach that the authors use is novel, well-thought through, and based on deep understanding of the complexity involved in regulation of hISC differentiation. Their work represents a building block for others working in this field to build upon and further advance the understanding of the role of EECs in the human intestine.

1. Minor – The authors briefly characterize other lineages in the enteroids under their conditions through qPCR under the different DM conditions (in supplemental data). It would be important to see IF for some other markers of intestinal lineages to see how their DM conditions affected the expression of other lineage markers at the protein level.

We appreciate this comment and have now included an analysis of immunofluorescent staining for CK20, MUC2 and LYZ for all conditions. We see staining of all three markers in the majority of the differentiation conditions, with G2D12 not showing any MUC2 staining and AS→RASP showing very little, if any, staining overall. We are very interested in further examining the role of these small molecules in the differentiation of other intestinal cell lineages in future studies. (Please see Supplementary Figures 7b-d)

2. Minor – In Figure 2, specify time point of photo for RSP enteroids

Thank you for pointing this out. The legend has been updated to reflect the time point as 14 days after starting the experiment.

3. Authors need to list all vendors or catalog numbers for many of the components that they use in their medium recipes such that others can reproduce their results.

Vendor and catalog numbers have now been included for each of the components used in the study. (Please see Supplementary Table 4)

Reviewers' Comments:

Reviewer #1:

Remarks to the Author:

The revised manuscript is considerably improved, and I commend the authors for attempting to address all my concerns and for the considerable effort employed at the experimental level. However, some concerns remain.

1. The novelty and the importance of the study as described is still misleading. There are several reports of small molecule use to direct EEC differentiation in human organoids including the use of BMP agonists, notch inhibitors, YAP inhibitors and ISX-9 and other combinations. In these manuscripts the effect of treatments is demonstrated at the transcriptional and protein level and in many cases include functional secretion studies.
2. The original protocols described to differentiate EEC's did advocate removal of wnt3a but there are many others that use reduced wnt3a eg. 15% conditioned media. This is important as some of the claimed benefit of the protocols described in the manuscript are based around the inadequacy of the current reported protocols abilities to maintain stem cell function.
3. The single cell sequencing experiments are successful in demonstrating increased EEC's and showing that these cells appear transcriptionally similar to native cell types. I would have liked to have seen this data used to identify if each treatment altered the trajectory of cells into specific EEC cell types. For example, we know that EEC's have two major branches of development peptide like and enterochromaffin like, then within the peptide EEC population several specific cell types can be delineated. Do the treatments increase all EEC subtypes or do they favour specific branches of development?

Reviewer #2:

Remarks to the Author:

The authors have been responsive to the previous review and the manuscript is improved by the addition of new data and new discussion.

- 1) The authors used rectoids to address applicability to other regions of the GI tract. They show that there are indeed different requirements for enteroendocrine induction/differentiation along the GI tract with rectoids responding well to media without additional small molecules. If the authors evaluated the effects of the small molecules on rectoids, it would be worthwhile to report, particularly if these small molecules altered the composition of the enteroendocrine cells present, i.e., do the small molecules "proximalize" the enteroendocrine cell population in rectoids.

The finding supports what other have shown, namely that stem cells from different regions maintain regional identity and would likely require different signals to direct regional-specific enteroendocrine populations.

- 2) The authors now more fully discuss the likelihood that effects of the putative GATA4 activator may indicate cell non-autonomous rather than autonomous effects or even that the activator works on related family members, i.e., GATA6, or even other yet unidentified targets.

A few minor tweaks to this discussion should be integrated:

- a) The statement that GATA4 is only expressed in enterocytes is not accurate. In addition to enterocytes, GATA4 is expressed in proliferating cells and Paneth cells along with enterocytes (Bosse et al, 2006). It is more accurate to state that GATA4 is absent in enteroendocrine and goblet cells rather than to state that it is only expressed in enterocytes.
- b) It is not quite clear what is meant by referring to CCK and GIP as known GATA4 targets, i.e., is it implied that these are direct transcriptional targets? Mouse work shows that GATA4 is absent in enteroendocrine cells so it is unlikely that, although levels of these markers change in the mutants, that these are direct GATA4 targets. It is possible that GATA4 expression in this experimental system differs from mouse but this hasn't been queried in the enteroids. It is more likely that regional identity changes in GATA4 mutants (GATA4 loss distalizes the proximal

intestine) underlie the changes in these targets expression, i.e, because population identities change. The work cited to support GATA4 as direct transcriptional regulator of the GIP promoter is purely in vitro work assessed by luciferase assays using 293T cells and EMSA. It doesn't hold up well against in vivo data.

3) It is somewhat counterintuitive that activating GATA4 would decrease enterocytes and increase secretory cells given effects observed in mouse mutants lacking GATA4, i.e, loss of GATA4 generally decreases enterocytes and increases secretory cells (scrNA-Seq data discussed on page 12).

4) Can the authors use their scrNA-Seq data to query expression of GATA4 in the treated enteroids to see if the small molecules ectopically induce GATA4 in the enteroendocrine lineages (and perhaps to validate cell type expression of GATA4 in their duodenal enteroids)?

Reviewer #3:

Remarks to the Author:

All my criticisms have been thoroughly addressed.

RESPONSE TO REVIEWER COMMENTS:

Reviewer #1 (Remarks to the Author):

The revised manuscript is considerably improved, and I commend the authors for attempting to address all my concerns and for the considerable effort employed at the experimental level.

However, some concerns remain.

We thank the reviewer for these encouraging comments and have sought to address the remaining comments raised below.

1. The novelty and the importance of the study as described is still misleading. There are several reports of small molecule use to direct EEC differentiation in human organoids including the use of BMP agonists, notch inhibitors, YAP inhibitors and ISX-9 and other combinations. In these manuscripts the effect of treatments is demonstrated at the transcriptional and protein level and in many cases include functional secretion studies.

We thank the reviewer for this important point and have now added multiple references that include RNA and protein studies. We have also indicated that, while not unique to our manuscript, these methods have been utilized relatively infrequently to date. (Please see changes on page 3-4, lines 76-81)

2. The original protocols described to differentiate EEC's did advocate removal of wnt3a but there are many others that use reduced wnt3a eg. 15% conditioned media. This is important as some of the claimed benefit of the protocols described in the manuscript are based around the inadequacy of the current reported protocols abilities to maintain stem cell function.

We thank the reviewer for this important point and apologize for this omission. We now reference manuscripts in the discussion that mention using reduced Wnt3a-conditioned media. (Please see changes on pages 19-20, lines 487-491)

3. The single cell sequencing experiments are successful in demonstrating increased EEC's and showing that these cells appear transcriptionally similar to native cell types. I would have liked to have seen this data used to identify if each treatment altered the trajectory of cells into specific EEC cell types. For example, we know that EEC's have two major branches of development *peptide like* and *enterochromaffin like*, then within the peptide EEC population several specific cell types can be delineated. Do the treatments increase all EEC subtypes or do they favour specific branches of development?

We appreciate this insightful comment and question. To understand the effect of each culture condition on the differentiation of individual enteroendocrine subsets, we further clustered the EE cells previously identified in our scRNA-seq dataset. We found that treatment of human organoids with RSP leads to a greater proportion of enteroendocrine progenitor cells compared with AS treatment and that treatment with AS leads to a greater proportion of mature EE cells compared with RSP. These mature cells are split into two groups, *TPH1*-expressing enterochromaffin cells and *ARX*-expressing non-enterochromaffin cells. Notably, AS treatment induced a higher proportion of enterochromaffin cells compared to RSP treatment. Moreover, while our dataset is under-powered to formally address the question, it appears that more hormone-producing M cells, defined by expression of *MLN*, are induced in response to treatment with RSP compared with AS treatment. (Please see changes on page 13, lines 315-333, pages 32-33, lines 812-824, and Supplementary Figure 5)

Reviewer #2 (Remarks to the Author):

The authors have been responsive to the previous review and the manuscript is improved by the addition of new data and new discussion.

We thank the reviewer for these positive comments.

1) The authors used rectoids to address applicability to other regions of the GI tract. They show that there are indeed different requirements for enteroendocrine induction/differentiation along the GI tract with rectoids responding well to media without additional small molecules. If the authors evaluated the effects of the small

molecules on rectoids, it would be worthwhile to report, particularly if these small molecules altered the composition of the enteroendocrine cells present, i.e., do the small molecules "proximalize" the enteroendocrine cell population in rectoids.

The finding supports what other have shown, namely that stem cells from different regions maintain regional identity and would likely require different signals to direct regional-specific enteroendocrine populations.

We agree with the reviewer's comment that our results reinforce the notion that regional identity is maintained *in vitro*. We also agree that it will be interesting to examine the various differentiation protocols with our rectoid cultures. Unfortunately, we have not yet evaluated these conditions with rectoids.

2) The authors now more fully discuss the likelihood that effects of the putative GATA4 activator may indicate cell non-autonomous rather than autonomous effects or even that the activator works on related family members, i.e., GATA6, or even other yet unidentified targets.

A few minor tweaks to this discussion should be integrated:

a) The statement that GATA4 is only expressed in enterocytes is not accurate. In addition to enterocytes, GATA4 is expressed in proliferating cells and Paneth cells along with enterocytes (Bosse et al, 2006). It is more accurate to state that GATA4 is absent in enteroendocrine and goblet cells rather than to state that it is only expressed in enterocytes.

b) It is not quite clear what is meant by referring to CCK and GIP as known GATA4 targets, i.e., is it implied that these are direct transcriptional targets? Mouse work shows that GATA4 is absent in enteroendocrine cells so it is unlikely that, although levels of these markers change in the mutants, that these are direct GATA4 targets. It is possible that GATA4 expression in this experimental system differs from mouse but this hasn't been queried in the enteroids. It is more likely that regional identity changes in GATA4 mutants (GATA4 loss distalizes the proximal intestine) underlie the changes in these targets expression, i.e, because population identities change. The work cited to support GATA4 as direct transcriptional regulator of the GIP promoter is purely *in vitro* work assessed by luciferase assays using 293T cells and EMSA. It doesn't hold up well against *in vivo* data.

We appreciate these insightful comments and apologize for any confusion that might have arisen by our discussion of GATA4. Based on current, and previous comments, we have changed our Introduction to focus on the endocannabinoid receptor signaling pathway, instead of GATA4, as a known activator of cells in the EE lineage. In our Discussion, we now hypothesize that GATA4 is a putative target of the cannabinoid receptor signaling pathway, highlighting both *in vitro* and *in vivo* evidence in support of a possible role for GATA4 in EE cell differentiation and/or function. (Please see changes on page 4, lines 90-93 and pages 20-21, lines 507-518)

3) It is somewhat counterintuitive that activating GATA4 would decrease enterocytes and increase secretory cells given effects observed in mouse mutants lacking GATA4, i.e, loss of GATA4 generally decreases enterocytes and increases secretory cells (scRNA-Seq data discussed on page 12).

We agree that this is counterintuitive, but, as indicated above, it is possible that the role of GATA4 in intestinal development and in EE cell function is different between mice and humans. Unfortunately, the additional studies required to examine the role of GATA4 in human intestinal cell development are beyond the scope of this study.

4) Can the authors use their scRNA-Seq data to query expression of GATA4 in the treated enteroids to see if the small molecules ectopically induce GATA4 in the enteroendocrine lineages (and perhaps to validate cell type expression of GATA4 in their duodenal enteroids)?

To further explore GATA4 expression in our human enteroid model, we examined our scRNA-Seq data and found GATA4 to be expressed in all cell types present in our dataset. In total, 35.8% of intestinal stem cells, 42.4% of proliferating progenitor cells, 23.5% of progenitor cells, 28.2% of enterocytes, 20.1% of goblet cells, and 24.1% of enteroendocrine cells had detectable GATA4 transcripts. This suggests a likely difference in the role of GATA4 in murine and human EE cells, which is beyond the scope of this manuscript.

Reviewer #3 (Remarks to the Author): All my criticisms have been thoroughly addressed.

Reviewers' Comments:

Reviewer #1:

Remarks to the Author:

The authors have addressed all my concerns.

Reviewer #2:

Remarks to the Author:

The authors have addressed all comments sufficiently.